# Truthful Data Acquisition via Peer Prediction

**Yiling Chen**
Harvard University
yiling@seas.harvard.edu

**Yiheng Shen**
Tsinghua University
shen-yh17@mails.tsinghua.edu.cn

**Shuran Zheng**
Harvard University
shuran_zheng@seas.harvard.edu

## Abstract

We consider the problem of purchasing data for machine learning or statistical estimation. The data analyst has a budget to purchase datasets from multiple data providers. She does not have any test data that can be used to evaluate the collected data and can assign payments to data providers solely based on the collected datasets. We consider the problem in the standard Bayesian paradigm and in two settings: (1) data are only collected once; (2) data are collected repeatedly and each day's data are drawn independently from the same distribution. For both settings, our mechanisms guarantee that truthfully reporting one's dataset is *always* an equilibrium by adopting techniques from peer prediction: pay each provider the *mutual information* between his reported data and other providers' reported data. Depending on the data distribution, the mechanisms can also discourage misreports that would lead to inaccurate predictions. Our mechanisms also guarantee individual rationality and budget feasibility for certain underlying distributions in the first setting and for all distributions in the second setting.

## 1  Introduction

Data has been the fuel of the success of machine learning and data science, which is becoming a major driving force for technological and economic growth. An important question is how to acquire high-quality data to enable learning and analysis when data are private possessions of data providers.

Naively, we could issue a constant payment to data providers in exchange for their data. But data providers can report more or less data than they actually have or even misreport values of their data without affecting their received payments. Alternatively, if we have a test dataset, we could reward data providers according to how well the model trained on their reported data performs on the test data. However, if the test dataset is biased, this could potentially incentivize data providers to bias their reported data toward the test set, which will limit the value of the acquired data for other learning or analysis tasks. Moreover, a test dataset may not even be available in many settings. In this work, we explore the design of reward mechanisms for acquiring high-quality data from multiple data providers when a data buyer doesn't have access to a test dataset. The ultimate goal is that, with the designed mechanisms, strategic data providers will find that truthfully reporting their possessed dataset is their best action and manipulation will lead to lower expected rewards. To make the mechanisms practical, we also require our mechanisms to always have non-negative and bounded payments so that data providers will find it beneficial to participate in (a.k.a. individual rationality) and the data buyer can afford the payments.

In a Bayesian paradigm where data are generated independently conditioned on some unknown parameters, we design mechanisms for two settings: (1) data are acquired only once, and (2) data

are acquired repeatedly and each day's data are independent from the previous days' data. For both settings, our mechanisms guarantee that truthfully reporting the datasets is always an equilibrium. For some models of data distributions, data providers in our mechanisms receive strictly lower rewards in expectation if their reported dataset leads to an inaccurate prediction of the underlying parameters, a property we called *sensitivity*.[1] While sensitivity doesn't strictly discourage manipulations of datasets that do not change the prediction of the parameters, it is a significant step toward achieving strict incentives for truthful reporting one's datasets, an ideal goal, especially because finding a manipulation without affecting the prediction of the parameters can be difficult. Our mechanisms guarantee IR and budget feasibility for certain underlying distributions in the first setting and for any underlying distributions in the second setting.

Our mechanisms are built upon recent developments [17, 16] in the peer prediction literature. The insight is that if we reward a data provider the mutual information [17] between his data and other providers' data, then by the data processing inequality, if other providers report their data truthfully, this data provider will only decrease the mutual information, hence his reward, by manipulating his dataset. We extend the peer prediction method developed by [16] to the data acquisition setting, and to further guarantee IR and budget feasibility. One of our major technical contributions is the explicit sensitivity guarantee of the peer-prediction style mechanisms, which is absent in the previous work.

## 2   Related Work

The problem of purchasing data from people has been investigated with different focuses, e.g. privacy concerns [14, 10, 13, 23, 7, 28], effort and cost of data providers[25, 4, 1, 5, 30, 6], reward allocation [12, 2]. Our work is the first to consider rewarding data without (good) test data that can be used evaluate the quality of reported data. Similar to our setting, [12, 2] consider paying to multiple data providers in a machine learning task. They use a test set to assess the contribution of subsets of data and then propose a fair measurement of the value of each data point in the dataset, which is based on the Shapley value in game theory. Both of the works do not formally consider the incentive compatibility of payment allocation. [28] proposes a market framework that purchases hypotheses for a machine learning problem when the data is distributed among multiple agents. Again they assume that the market has access to some true samples and the participants are paid with their incremental contributions evaluated by these true samples. Besides, there is a small literature (see [9] and subsequent work) on aggregating datasets using scoring rules that also considers signal distributions in exponential families.

The main techniques of this work come from the literature of *peer prediction* [19, 24, 8, 11, 26, 17, 16, 18, 15]. Peer prediction is the problem of information elicitation without verification. The participants receive correlated signals of an unknown ground truth and the goal is to elicit the true signals from the participants. In our problem, the dataset can be viewed as a signal of the ground truth. What makes our problem more challenging than the standard peer prediction problem is that (1) the signal space is much larger and (2) the correlation between signals is more complicated. Standard peer prediction mechanisms either require the full knowledge of the underlying signal distribution, or make assumptions on the signal distribution that are not applicable to our problem. [16] applies the peer prediction method to the co-training problem, in which two participants are asked to submit forecasts of latent labels in a machine learning problem. Our work is built upon the main insights of [16]. We discuss the differences between our model and theirs in the model section, and show how their techniques are applied in the result sections.

Our work is also related to Multi-view Learning (see [29] for a survey). But our work focuses on the data acquisition, but not the machine learning methods used on the (multi-view) data.

## 3   Model

A data analyst wants to gather data for some future statistical estimation or machine learning tasks. There are $n$ data providers. The $i$-th data provider holds a dataset $D_i$ consisting of $N_i$ data points $d_i^{(1)}, \ldots, d_i^{(N_i)}$ with support $\mathcal{D}_i$. The data generation follows a standard Bayesian process. For each

data set $D_i$, data points $d_i^{(j)} \in D_i$ are i.i.d. samples conditioned on some unknown parameters $\boldsymbol{\theta} \in \Theta$. Let $p(\boldsymbol{\theta}, D_1, \dots, D_n)$ be the joint distribution of $\boldsymbol{\theta}$ and $n$ data providers' datasets. We consider two types of spaces for $\Theta$ in this paper: (1) $\boldsymbol{\theta}$ has finite support, i.e., $|\Theta| = m$ is finite, and (2) $\boldsymbol{\theta}$ has continuous support, i.e. $\boldsymbol{\theta} \in \mathbb{R}^m$ and $\Theta \subseteq \mathbb{R}^m$. For the case of continuous support, to alleviate computational issues, we consider a widely used class of distributions, an *exponential family*.

The data analyst's goal is to incentivize the data providers to give their true datasets with a budget $B$. She needs to design a payment rule $r_i(\widetilde{D}_1, \dots, \widetilde{D}_n)$ for $i \in [n]$ that decides how much to pay data provider $i$ according to all the reported datasets $\widetilde{D}_1, \dots, \widetilde{D}_n$. The payment rule should ideally incentivize truthful reporting, that is, $\widetilde{D}_i = D_i$ for all $i$.

Before we formally define the desirable properties of a payment rule, we note that the analyst will have to leverage the correlation between people's data to distinguish a misreported dataset from a true dataset because all she has access to is the reported datasets. To make the problem tractable, we thus make the following assumption about the data correlation: parameters $\boldsymbol{\theta}$ contains all the mutual information between the datasets. More formally, the datasets are independent conditioned on $\boldsymbol{\theta}$.

**Assumption 3.1.** $D_1, \dots, D_n$ *are independent conditioned on* $\boldsymbol{\theta}$,

$$p(D_1, \dots, D_n | \boldsymbol{\theta}) = p(D_1 | \boldsymbol{\theta}) \cdots p(D_n | \boldsymbol{\theta}).$$

This is definitely not an assumption that would hold for arbitrarily picked parameters $\boldsymbol{\theta}$ and any datasets. One can easily find cases where the datasets are correlated to some parameters other than $\boldsymbol{\theta}$. So the data analyst needs to carefully decide what to include in $\boldsymbol{\theta}$ and $D_i$, by either expanding $\boldsymbol{\theta}$ to include all relevant parameters or reducing the content of $D_i$ to exclude all redundant data entries that can cause extra correlations.

**Example 3.1.** *Consider the linear regression model where provider $i$'s data points $d_i^{(j)} = (\mathbf{z}_i^{(j)}, y_i^{(j)})$ consist of a feature vector $\mathbf{z}_i^{(j)}$ and a label $y_i^{(j)}$. We have a linear model*

$$y_i^{(j)} = \boldsymbol{\theta}^T \mathbf{z}_i^{(j)} + \varepsilon_i^{(j)}.$$

*Then datasets $D_1, \dots, D_n$ will be independent conditioning on $\boldsymbol{\theta}$ as long as (1) different data providers draw their feature vectors independently, i.e., $\mathbf{z}_1^{(j_1)}, \dots, \mathbf{z}_n^{(j_n)}$ are independent for all $j_1 \in [N_1], \dots, j_n \in [N_n]$, and (2) the noises are independent.*

We further assume that the data analyst has some insight about the data generation process.

**Assumption 3.2.** *The data analyst possesses a commonly accepted prior $p(\boldsymbol{\theta})$ and a commonly accepted model for data generating process so that she can compute the posterior $p(\boldsymbol{\theta}|D_i), \forall i, D_i$.*

When $|\Theta|$ is finite, $p(\boldsymbol{\theta}|D_i)$ can be computed as a function of $p(\boldsymbol{\theta}|d_i)$ using the method in Appendix B. For a model in the exponential family, $p(\boldsymbol{\theta}|D_i)$ can be computed as in Definition 4.2.

Note that we do not always require the data analyst to know the whole distribution $p(D_i|\boldsymbol{\theta})$, it suffices for the data analyst to have the necessary information to compute $p(\boldsymbol{\theta}|D_i)$.

**Example 3.2.** *Consider the linear regression model in Example 3.1. We use $\mathbf{z}_i$ to represent all the features in $D_i$ and use $\mathbf{y}_i$ to represent all the labels in $D_i$. If the features $\mathbf{z}_i$ are independent from $\boldsymbol{\theta}$, the data analyst does not need to know the distribution of $\mathbf{z}_i$. It suffices to know $p(\mathbf{y}_i|\mathbf{z}_i, \boldsymbol{\theta})$ and $p(\boldsymbol{\theta})$ to know $p(\boldsymbol{\theta}|D_i)$ because*

$$p(\boldsymbol{\theta}|(\mathbf{z}_i, \mathbf{y}_i)) \propto p((\mathbf{z}_i, \mathbf{y}_i)|\boldsymbol{\theta})p(\boldsymbol{\theta}) = p(\mathbf{y}_i|\mathbf{z}_i, \boldsymbol{\theta})p(\mathbf{z}_i|\boldsymbol{\theta})p(\boldsymbol{\theta}) = p(\mathbf{y}_i|\mathbf{z}_i, \boldsymbol{\theta})p(\mathbf{z}_i)p(\boldsymbol{\theta})$$
$$\propto p(\mathbf{y}_i|\mathbf{z}_i, \boldsymbol{\theta})p(\boldsymbol{\theta}).$$

Finally we assume that the identities of the providers can be verified.

**Assumption 3.3.** *The data analyst can verify the data providers' identities, so one data provider can only submit one dataset and get one payment.*

We now formally introduce some desirable properties of a payment rule. We say that a payment rule is *truthful* if reporting true datasets is a weak equilibrium, that is, when the others report true datasets, it is also (weakly) optimal for me to report the true dataset (based on my own belief).

**Definition 3.1** (Truthfulness). *Let $D_{-i}$ be the datasets of all providers except $i$. A payment rule $\mathbf{r}(D_1, \ldots, D_n)$ is truthful if: for any (commonly accepted model of) underlying distribution $p(\boldsymbol{\theta}, D_1, \ldots, D_n)$, for every data provider $i$ and any realization of his dataset $D_i$, when all other data providers truthfully report $D_{-i}$, truthfully reporting $D_i$ leads to the highest expected payment, where the expectation is taken over the distribution of $D_{-i}$ conditioned on $D_i$, i.e.,*

$$\mathbb{E}_{D_{-i} \sim p(D_{-i}|D_i)}[r_i(D_i, D_{-i})] \geq \mathbb{E}_{D_{-i} \sim p(D_{-i}|D_i)}[r_i(D_i', D_{-i})], \quad \forall i, D_i, D_i'.$$

Note that this definition does not require the agents to actually know the conditional distribution and to be able to evaluate the expectation themselves. It is a guarantee that no matter what the underlying distribution is, truthfully reporting is an equilibrium.

Because truthfulness is defined as a weak equilibrium, it does not necessarily discourage misreporting.[2] What it ensures is that the mechanism does not encourage misreporting.[3] So, we want a stronger guarantee than truthfulness. We thus define *sensitivity*: the expected payment should be strictly lower when the reported data does not give the accurate prediction of $\boldsymbol{\theta}$.

**Definition 3.2** (Sensitivity). *A payment rule $\mathbf{r}(D_1, \ldots, D_n)$ is sensitive if for any (commonly accepted model of) underlying distribution $p(\boldsymbol{\theta}, D_1, \ldots, D_n)$, for any provider $i$ and any realization of his dataset $D_i$, when all other providers $j \neq i$ report $\widetilde{D}_j(D_j)$ with accurate posterior $p(\boldsymbol{\theta}|\widetilde{D}_j(D_j)) = p(\boldsymbol{\theta}|D_j)$, we have (1) truthfully reporting $D_i$ leads to the highest expected payment*

$$\mathbb{E}_{D_{-i} \sim p(D_{-i}|D_i)}[r_i(D_i, \widetilde{D}_{-i}(D_{-i}))] \geq \mathbb{E}_{D_{-i} \sim p(D_{-i}|D_i)}[r_i(D_i', \widetilde{D}_{-i}(D_{-i}))], \; \forall D_i'$$

*and (2) reporting a dataset $D_i'$ with inaccurate posterior $p(\boldsymbol{\theta}|D_i') \neq p(\boldsymbol{\theta}|D_i)$ is strictly worse than reporting a dataset $\widetilde{D}_i$ with accurate posterior $p(\boldsymbol{\theta}|\widetilde{D}_i) = p(\boldsymbol{\theta}|D_i)$,*

$$\mathbb{E}_{D_{-i} \sim p(D_{-i}|D_i)}[r_i(\widetilde{D}_i, \widetilde{D}_{-i}(D_{-i}))] > \mathbb{E}_{D_{-i} \sim p(D_{-i}|D_i)}[r_i(D_i', \widetilde{D}_{-i}(D_{-i}))],$$

*Furthermore, let $\Delta_i = p(\boldsymbol{\theta}|D_i') - p(\boldsymbol{\theta}|D_i)$, a payment rule is $\alpha$-sensitive for agent $i$ if*

$$\mathbb{E}_{D_{-i} \sim p(D_{-i}|D_i)}[r_i(D_i, \widetilde{D}_{-i}(D_{-i}))] - \mathbb{E}_{D_{-i} \sim p(D_{-i}|D_i)}[r_i(D_i', \widetilde{D}_{-i}(D_{-i}))] \geq \alpha\|\Delta_i\|,$$

*for all $D_i, D_i'$ and reports $\widetilde{D}_{-i}(D_{-i})$ that give the accurate posteriors.*

Our definition of sensitivity guarantees that at an equilibrium, the reported datasets must give the correct posteriors $p(\boldsymbol{\theta}|\widetilde{D}_i) = p(\boldsymbol{\theta}|D_i)$. We can further show that at an equilibrium, the analyst will get the accurate posterior $p(\boldsymbol{\theta}|D_1, \ldots, D_n)$.

**Lemma 3.1.** *When $D_1, \ldots, D_n$ are independent conditioned on $\boldsymbol{\theta}$, for any $(D_1, \ldots, D_n)$ and $(\widetilde{D}_1, \ldots, \widetilde{D}_n)$, if $p(\boldsymbol{\theta}|D_i) = p(\boldsymbol{\theta}|\widetilde{D}_i) \; \forall i$, then $p(\boldsymbol{\theta}|D_1, \ldots, D_n) = p(\boldsymbol{\theta}|\widetilde{D}_1, \ldots, \widetilde{D}_n)$.*

A more ideal property would be that the expected payment is strictly lower for any dataset $D_i' \neq D_i$. Mechanisms that satisfy sensitivity can be viewed as an important step toward this ideal goal, as the only possible payment-maximizing manipulations are to report a dataset $\widetilde{D}_i$ that has the correct posterior $p(\boldsymbol{\theta}|\widetilde{D}_i) = p(\boldsymbol{\theta}|D_i)$. Arguably, finding such a manipulation can be challenging. Sensitivity guarantees the accurate prediction of $\boldsymbol{\theta}$ at an equilibrium.

Second, we want a fair payment rule that is indifferent to data providers' identities.

**Definition 3.3** (Symmetry). *A payment rule $r$ is symmetric if for all permutation of $n$ elements $\pi(\cdot)$, $r_i(D_1, \ldots, D_n) = r_{\pi(i)}(D_{\pi(1)}, \ldots, D_{\pi(n)})$ for all $i$.*

Third, we want non-negative payments and the total payment should not exceed the budget.

**Definition 3.4** (Individual rationality and budget feasibility). *A payment rule $r$ is individually rational if $r_i(D_1, \ldots, D_n) \geq 0, \quad \forall i, D_1, \ldots, D_n$. A payment rule $r$ is budget feasible if $\sum_{i=1}^n r_i(D_1, \ldots, D_n) \leq B, \forall D_1, \ldots, D_n$.*

We will consider two acquisition settings in this paper:

**One-time data acquisition.** The data analyst collects data in one batch. In this case, our problem is very similar to the single-task forecast elicitation in [16]. But our model considers the budget feasibility and the IR, whereas they only consider the truthfulness of the mechanism.

**Multiple-time data acquisition.** The data analyst repeatedly collects data for $T \geq 2$ days. On day $t$, $(\boldsymbol{\theta}^{(t)}, D_1^{(t)}, \ldots, D_n^{(t)})$ is drawn independently from the same distribution $p(\boldsymbol{\theta}, D_1, \ldots, D_n)$. The analyst has a budget $B^{(t)}$ and wants to know the posterior of $\boldsymbol{\theta}^{(t)}$, $p(\boldsymbol{\theta}^{(t)}|D_1^{(t)}, \ldots, D_n^{(t)})$. In this case, our setting differs from the multi-task forecast elicitation in [16] because providers can decide their strategies on a day based on all the observed historical data before that day.[4] The multi-task forecast elicitation in [16] asks the agents to submit forecasts of latent labels in multiple similar independent tasks. It is assumed that the agent's forecast strategy for one task only depends on his information about that task but not the information about other tasks.

# 4 Preliminaries

In this section, we introduce some necessary background for developing our mechanisms. We first give the definitions of *exponential family* distributions. Our designed mechanism will leverage the idea of mutual information between reported datasets to incentivize truthful reporting.

## 4.1 Exponential Family

**Definition 4.1** (Exponential family [21]). *A likelihood function $p(\mathbf{x}|\boldsymbol{\theta})$, for $\mathbf{x} = (x_1, \ldots, x_n) \in \mathcal{X}^n$ and $\boldsymbol{\theta} \in \Theta \subseteq \mathbb{R}^m$ is said to be in the* exponential family *in canonical form if it is of the form*

$$p(\mathbf{x}|\boldsymbol{\theta}) = \frac{1}{Z(\boldsymbol{\theta})} h(\mathbf{x}) \exp\left[\boldsymbol{\theta}^T \boldsymbol{\phi}(\mathbf{x})\right] \quad or \quad p(\mathbf{x}|\boldsymbol{\theta}) = h(\mathbf{x}) \exp\left[\boldsymbol{\theta}^T \boldsymbol{\phi}(\mathbf{x}) - A(\boldsymbol{\theta})\right] \quad (1)$$

*Here $\boldsymbol{\phi}(x) \in \mathbb{R}^m$ is called a vector of* sufficient statistics, *$Z(\boldsymbol{\theta}) = \int_{\mathcal{X}^n} h(\mathbf{x}) \exp\left[\boldsymbol{\theta}^T \boldsymbol{\phi}(\mathbf{x})\right]$ is called the* partition function, *$A(\boldsymbol{\theta}) = \ln Z(\boldsymbol{\theta})$ is called the* log partition function.

In Bayesian probability theory, if the posterior distributions $p(\boldsymbol{\theta}|\mathbf{x})$ are in the same probability distribution family as the prior probability distribution $p(\boldsymbol{\theta})$, the prior and posterior are then called conjugate distributions, and the prior is called a conjugate prior for the likelihood function.

**Definition 4.2** (Conjugate prior for the exponential family [21]). *For a likelihood function in the exponential family $p(\mathbf{x}|\boldsymbol{\theta}) = h(\mathbf{x}) \exp\left[\boldsymbol{\theta}^T \boldsymbol{\phi}(\mathbf{x}) - A(\boldsymbol{\theta})\right]$. The conjugate prior for $\boldsymbol{\theta}$ with parameters $\nu_0, \overline{\boldsymbol{\tau}}_0$ is of the form*

$$p(\boldsymbol{\theta}) = \mathcal{P}(\boldsymbol{\theta}|\nu_0, \overline{\boldsymbol{\tau}}_0) = g(\nu_0, \overline{\boldsymbol{\tau}}_0) \exp\left[\nu_0 \boldsymbol{\theta}^T \overline{\boldsymbol{\tau}}_0 - \nu_0 A(\boldsymbol{\theta})\right]. \quad (2)$$

*Let $\overline{\boldsymbol{s}} = \frac{1}{n} \sum_{i=1}^{n} \boldsymbol{\phi}(x_i)$. Then the posterior of $\boldsymbol{\theta}$ can be represented in the same form as the prior*

$$p(\boldsymbol{\theta}|\mathbf{x}) \propto \exp\left[\boldsymbol{\theta}^T (\nu_0 \overline{\boldsymbol{\tau}}_0 + n\overline{\boldsymbol{s}}) - (\nu_0 + n) A(\boldsymbol{\theta})\right] = \mathcal{P}\left(\boldsymbol{\theta}\middle|\nu_0 + n, \frac{\nu_0 \overline{\boldsymbol{\tau}}_0 + n\overline{\boldsymbol{s}}}{\nu_0 + n}\right),$$

*where $\mathcal{P}\left(\boldsymbol{\theta}|\nu_0 + n, \frac{\nu_0 \overline{\boldsymbol{\tau}}_0 + n\overline{\boldsymbol{s}}}{\nu_0 + n}\right)$ is the conjugate prior with parameters $\nu_0 + n$ and $\frac{\nu_0 \overline{\boldsymbol{\tau}}_0 + n\overline{\boldsymbol{s}}}{\nu_0 + n}$.*

A lot of commonly used distributions belong to the exponential family. Gaussian, Multinoulli, Multinomial, Geometric, etc. Due to the space limit, we introduce only the definitions and refer the readers who are not familiar with the exponential family to [21] for more details.

## 4.2 Mutual Information

We will use the *point-wise mutual information* and the *$f$-mutual information gain* defined in [16]. We introduce this notion of mutual information in the context of our problem.

**Definition 4.3** (Point-wise mutual information). *We define the point-wise mutual information between two datasets $D_1$ and $D_2$ to be*

$$PMI(D_1, D_2) = \int_{\boldsymbol{\theta} \in \Theta} \frac{p(\boldsymbol{\theta}|D_1)p(\boldsymbol{\theta}|D_2)}{p(\boldsymbol{\theta})} \, d\boldsymbol{\theta}. \tag{3}$$

For finite case, we define $PMI(D_1, D_2) = \sum_{\boldsymbol{\theta} \in \Theta} \frac{p(\boldsymbol{\theta}|D_1)p(\boldsymbol{\theta}|D_2)}{p(\boldsymbol{\theta})} \, d\boldsymbol{\theta}$.

When $|\Theta|$ is finite or a model in the exponential family is used, the PMI will be computable.

**Lemma 4.1.** *When $|\Theta|$ is finite, $PMI(\cdot)$ can be computed in $O(|\Theta|)$ time. If a model in exponential family is used, so that the prior and all the posterior of $\boldsymbol{\theta}$ can be written in the form*

$$p(\boldsymbol{\theta}) = \mathcal{P}(\boldsymbol{\theta}|\nu_0, \overline{\boldsymbol{\tau}}_0) = g(\nu_0, \overline{\boldsymbol{\tau}}_0) \exp\left[\nu_0 \boldsymbol{\theta}^T \overline{\boldsymbol{\tau}}_0 - \nu_0 A(\boldsymbol{\theta})\right],$$

*$p(\boldsymbol{\theta}|D_i) = \mathcal{P}(\boldsymbol{\theta}|\nu_i, \overline{\boldsymbol{\tau}}_i)$ and $p(\boldsymbol{\theta}|D_{-i}) = \mathcal{P}(\boldsymbol{\theta}|\nu_{-i}, \overline{\boldsymbol{\tau}}_{-i})$, then the point-wise mutual information can be computed as*

$$PMI(D_i, D_{-i}) = \frac{g(\nu_i, \overline{\boldsymbol{\tau}}_i)g(\nu_{-i}, \overline{\boldsymbol{\tau}}_{-i})}{g(\nu_0, \overline{\boldsymbol{\tau}}_0)g(\nu_i + \nu_{-i} - \nu_0, \frac{\nu_i \overline{\boldsymbol{\tau}}_i + \nu_{-i} \overline{\boldsymbol{\tau}}_{-i} - \nu_0 \overline{\boldsymbol{\tau}}_0}{\nu_i + \nu_{-i} - \nu_0})}.$$

For single-task forecast elicitation, [16] proposes a truthful payment rule.

**Definition 4.4** (log-PMI payment [16]). *Suppose there are two data providers reporting $\widetilde{D}_A$ and $\widetilde{D}_B$ respectively. Then the* log-*PMI rule pays them $r_A = r_B = \log(PMI(\widetilde{D}_A, \widetilde{D}_B))$.*

**Proposition 4.1.** *When the* log-*PMI rule is used, the expected payment equals the mutual information between $\widetilde{D}_A$ and $\widetilde{D}_B$, where the expectation is taken over the distribution of $\widetilde{D}_A$ and $\widetilde{D}_B$.*

We now give the definition of the $f$-mutual information. An $f$-divergence is a function that measures the difference between two probability distributions.

**Definition 4.5** ($f$-divergence). *Given a convex function $f$ with $f(1) = 0$, for two distributions over $\Omega$, $p, q \in \Delta\Omega$, define the $f$-divergence of $p$ and $q$ to be $D_f(p, q) = \int_{\omega \in \Omega} p(\omega) f\left(\frac{q(\omega)}{p(\omega)}\right)$.*

The $f$-*mutual information* of two random variables is a measure of the mutual dependence of two random variables, which is defined as the $f$-divergence between their joint distribution and the product of their marginal distributions.

In duality theory, the convex conjugate of a function is defined as follows.

**Definition 4.6** (Convex conjugate). *For any function $f : \mathbb{R} \to \mathbb{R}$, define the convex conjugate function of $f$ as $f^*(y) = \sup_x xy - f(x)$.*

The following inequality ([22, 16]) will be used in our proof.

**Lemma 4.2** (Lemma 1 in [22]). *For any differentiable convex function $f$ with $f(1) = 0$, any two distributions over $\Omega$, $p, q \in \Delta\Omega$, let $\mathcal{G}$ be the set of all functions from $\Omega$ to $\mathbb{R}$, then we have*

$$D_f(p, q) \geq \sup_{g \in \mathcal{G}} \int_{\omega \in \Omega} g(\omega)q(\omega) - f^*(g(\omega))p(\omega) \, d\omega = \sup_{g \in \mathcal{G}} \mathbb{E}_q g - \mathbb{E}_p f^*(g).$$

*A function $g$ achieves equality if and only if $g(\omega) \in \partial f\left(\frac{q(\omega)}{p(\omega)}\right) \forall \omega$ with $p(\omega) > 0$, where $\partial f\left(\frac{q(\omega)}{p(\omega)}\right)$ represents the subdifferential of $f$ at point $q(\omega)/p(\omega)$.*

## 5 One-time Data Acquisition

In this section we apply [16]'s log-PMI payment rule to our one-time data acquisition problem. The log-PMI payment rule ensures truthfulness, but its payment can be negative or unbounded or even ill-defined. So we mainly focus on the mechanism's sensitivity, budget feasibility and IR. To guarantee budget feasibility and IR, our mechanism requires a lower bound and an upper bound of PMI, which may be difficult to find for some models in the exponential family.

If the analyst knows the distribution $p(D_i|\boldsymbol{\theta})$, then she will be able to compute $p(D_{-i}|D_i) = \sum_{\boldsymbol{\theta}} p(D_{-i}|\boldsymbol{\theta})p(\boldsymbol{\theta}|D_i)$. In this case, we can employ peer prediction mechanisms [19] to design payments and guarantee truthfulness. In Appendix C.1, we give an example of such mechanisms.

In this work we do not assume that $p(D_i|\boldsymbol{\theta})$ is known (see Example 3.2). When $p(D_i|\boldsymbol{\theta})$ is unknown but the analyst can compute $p(\boldsymbol{\theta}|D_i)$, our idea is to use the log-PMI payment rule in [16] and then add a normalization step to ensure budget feasibility and IR. However the log-PMI will be ill-defined if $PMI = 0$. To avoid this, for each possible $D_{-i}$, we define set $\mathbb{D}_i(D_{-i}) = \{D_i|PMI(D_i, D_{-i}) > 0\}$ and the log-PMI will only be computed for $\widetilde{D}_i \in \mathbb{D}_i(\widetilde{D}_{-i})$.

The normalization step will require an upper bound $R$ and lower bound $L$ of the log-PMI payment.[5] If $|\Theta|$ is finite, we can find a lower bound and an upper bound in polynomial time, which we prove in Appendix C.2. When a model in the exponential family is used, it is more difficult to find $L$ and $R$. By Lemma 4.1, if the $g$ function is bounded, we will be able to bound the payment. For example, if we are estimating the mean of a univariate Gaussian with known variance, $L$ and $R$ will be bounded if the number of data points is bounded. Details can be found in Appendix C.3. Our mechanism works as follows.

---

**Mechanism 1:** One-time data collecting mechanism.

---

(1) Ask all data providers to report their datasets $\widetilde{D}_1, \ldots, \widetilde{D}_n$.
(2) If data provider $i$'s reported dataset $\widetilde{D}_i \in \mathbb{D}_i(\widetilde{D}_{-i})$, we compute a score for his dataset
$s_i = \log PMI(\widetilde{D}_i, \widetilde{D}_{-i})$.
(3) The final payment for data provider $i$ is:

$$r_i(\widetilde{D}_1, \ldots, \widetilde{D}_n) = \begin{cases} \frac{B}{n} \cdot \frac{s_i - L}{R - L} & \text{if } \widetilde{D}_i \in \mathbb{D}_i(\widetilde{D}_{-i}) \\ 0 & \text{otherwise.} \end{cases}$$

---

**Theorem 5.1.** *Mechanism 1 is IR, truthful, budget feasible, symmetric.*

Note that by Proposition 4.1, the expected payment for a data provider is decided by the mutual information between his data and other people's data. The payments are efficiently computable for finite-size $\Theta$ and for models in exponential family (Lemma 4.1).

Next, we discuss the sensitivity. In [16], checking whether the mechanism will be sensitive requires knowing whether a system of linear equations (which has an exponential size in our problem) has a unique solution. So it is not clear how likely the mechanisms will be sensitive. In our data acquisition setting, we are able to give much stronger and more explicit guarantees. This kind of stronger guarantee is possible because of the special structure of the reports (or the signals) that each dataset consists of i.i.d. samples.

We first define some notations. When $|\Theta|$ is finite, let $\boldsymbol{Q_{-i}}$ be a $(\Pi_{j\in[n],j\neq i}|\mathcal{D}_j|^{N_j}) \times |\Theta|$ matrix that represents the conditional distribution of $\boldsymbol{\theta}$ conditioning on every realization of $D_{-i}$. So the element in row $D_{-i}$ and column $\boldsymbol{\theta}$ is equal to $p(\boldsymbol{\theta}|D_{-i})$. We also define the data generating matrix $G_i$ with $|\mathcal{D}_i|$ rows and $|\Theta|$ columns. Each row corresponds to a possible data point $d_i \in \mathcal{D}_i$ in the dataset and each column corresponds to a $\boldsymbol{\theta} \in \Theta$. The element in the row corresponding to data point $d_i$ and the column $\boldsymbol{\theta}$ is $p(\boldsymbol{\theta}|d_i)$. We give the sufficient condition for the mechanism to be sensitive.

**Theorem 5.2.** *When $|\Theta|$ is finite, Mechanism 1 is sensitive if for all $i$, $Q_{-i}$ has rank $|\Theta|$.*

Since the size of $Q_{-i}$ can be exponentially large, it may be computationally infeasible to check the rank of $Q_{-i}$. We thus give a simpler condition that only uses $G_i$, which has a polynomial size.

**Definition 5.1.** *The Kruskal rank (or $k$-rank) of a matrix $M$, denoted by $rank_k(M)$, is the maximal number $r$ such that any set of $r$ columns of $M$ is linearly independent.*

**Corollary 5.1.** *When $|\Theta|$ is finite, Mechanism 1 is sensitive if for all $i$, $\sum_{j\neq i} (rank_k(G_j) - 1) \cdot N_j + 1 \geq |\Theta|$, where $N_j$ is the number of data points in $D_j$.*

In Appendix C.5.1, we also give a lower bound for $\alpha$ so that Mechanism 1 is $\alpha$-sensitive.

Our sensitivity results (Theorem 5.2 and Corollary 5.1) basically mean that when there is enough correlation between other people's data $D_{-i}$ and $\theta$, the mechanism will be sensitive. Corollary 5.1

quantifies the correlation using the k-rank of the data generating matrix. It is arguably not difficult to have enough correlation: a naive relaxation of Corollary 5.1 says that assuming different $\theta$ lead to different data distributions (so that $rank_k(G_j) \geq 2$), the mechanism will be sensitive if for any provider $i$, the total number of other people's data points $\geq |\Theta| - 1$.

When $\Theta \subseteq \mathbb{R}^m$, it becomes more difficult to guarantee sensitivity. Suppose the data analyst uses a model from the exponential family so that the prior and all the posterior of $\theta$ can be written in the form in Lemma 4.1. The sensitivity of the mechanism will depend on the normalization term $g(\nu, \overline{\tau})$ (or equivalently, the partition function) of the pdf. More specifically, define

$$h_{D_{-i}}(\nu_i, \overline{\tau}_i) = \frac{g(\nu_i, \overline{\tau}_i)}{g(\nu_i + \nu_{-i} - \nu_0, \frac{\nu_i \overline{\tau}_i + \nu_{-i} \overline{\tau}_{-i} - \nu_0 \overline{\tau}_0}{\nu_i + \nu_{-i} - \nu_0})}, \tag{4}$$

then we have the following sufficient and necessary conditions for the sensitivity of the mechanism.

**Theorem 5.3.** *When* $\Theta \subseteq \mathbb{R}^m$, *if the data analyst uses a model in the exponential family, then Mechanism 1 is sensitive if and only if for any* $(\nu_i', \overline{\tau}_i') \neq (\nu_i, \overline{\tau}_i)$, *we have* $\mathrm{Pr}_{D_{-i}}[h_{D_{-i}}(\nu_i', \overline{\tau}_i') \neq h_{D_{-i}}(\nu_i, \overline{\tau}_i)] > 0$.

The theorem basically means that the mechanism will be sensitive if any pairs of different reports that will lead to different posteriors of $\theta$ can be distinguished by $h_{D_{-i}}(\cdot)$ with non-zero probability. However, for different models in the exponential family, this is not always true. For example, if we estimate the mean $\mu$ of a univariate Gaussian with a known variance and the Gaussian conjugate prior is used, then the normalization term only depends on the variance but not the mean, so in this case $h(\cdot)$ can only detect the change in variance, which means that the mechanism will be sensitive to replication and withholding, but not necessarily other types of manipulations. But if we estimate the mean of a Bernoulli distribution whose conjugate prior is the Beta distribution, then the partition function will be the Beta function, which can detect different posteriors and thus the mechanism will be sensitive. See Appendix C.4 for more details. The missing proofs can be found in Appendix C.5.

# 6   Multiple-time Data Acquisition

Now we consider the case when the data analyst needs to repeatedly collect data for the same task. At day $t$, the analyst has a budget $B^{(t)}$ and a new ensemble $(\theta^{(t)}, D_1^{(t)}, \ldots, D_n^{(t)})$ is drawn from the same distribution $p(\theta, D_1, \ldots, D_n)$, independent of the previous data. Again we assume that the data generating distribution $p(D_i|\theta)$ can be unknown, but the analyst is able to compute $p(\theta|D_i)$) after seeing the data(See Example 3.2). The data analyst can use the one-time purchasing mechanism (Section 5) at each round. But we show that if the data analyst can give the payment one day after the data is reported, a broader class of mechanisms can be applied to guarantee the desirable properties, which ensures bounded payments without any assumptions on the underlying distribution. Our method is based on the *f-mutual information gain* in [16] for multi-task forecast elicitation. The payment function in [16] has a minor error. We correct the payment function in this work.[6]

Our mechanism (Mechanism 2) works as follows. On day $t$, the data providers are first asked to report their data for day $t$. Then for each provider $i$, we use the other providers' reported data on day $t-1$ and day $t$ to evaluate provider $i$'s reported data on day $t-1$. A score $s_i$ will be computed for each provider's $\widetilde{D}_i^{(t-1)}$. The score $s_i$ is defined in the same way as the *f-mutual information gain* in [16], which is specified by a differentiable convex function $f : \mathbb{R} \to \mathbb{R}$ and its convex conjugate $f^*$ (Definition 4.6),

$$s_i = f'\left(\frac{1}{PMI(\widetilde{D}_i^{(t-1)}, \widetilde{D}_{-i}^{(t)})}\right) - f^*\left(f'\left(\frac{1}{PMI(\widetilde{D}_i^{(t-1)}, \widetilde{D}_{-i}^{(t-1)})}\right)\right).^{[7]} \tag{5}$$

The score is defined in this particular way because it can guarantee truthfulness according to Lemma 4.2. It can be proved that when the agents truthfully report $D_i$, the expectation of $s_i$ will reach the supremum in Lemma 4.2, and will then be equal to the $f$-mutual information of $D_i^{(t-1)}$

and $D_{-i}^{(t-1)}$. We can further prove that if data provider $i$ reports a dataset $\widetilde{D}_i^{(t-1)}$ that leads to a different posterior $p(\boldsymbol{\theta}|\widetilde{D}_i^{(t-1)}) \neq p(\boldsymbol{\theta}|D_i^{(t-1)})$, the expectation of $s_i$ will deviate from the supremum in Lemma 4.2 and thus get lower.

According to the definition (5), if we carefully choose the convex function $f$ to be a differentiable convex function with a bounded derivative $f' \in [0, U]$ and with the convex conjugate $f^*$ bounded on $[0, U]$, then the scores $s_1, \ldots, s_n$ will always be bounded. We can then normalize $s_1, \ldots, s_n$ so that the payments are non-negative and the total payment does not exceed $B^{(t-1)}$. Here we give one possible choice of $f'$ that can guarantee bounded scores: the Logistic function $\frac{1}{1+e^{-x}}$.

| $f(x)$ | $f'(x)$ | range of $f'(x)$ | $f^*(x)$ | range of $f^*(x)$ |
|---|---|---|---|---|
| $\ln(1+e^x)$ | $\frac{1}{1+e^{-x}}$ | $[\frac{1}{2}, 1)$ on $\mathbb{R}^{\geq 0}$ | $x \ln x + (1-x)\ln(1-x)$ | $[-\ln 2, 0]$ on $[\frac{1}{2}, 1)$ |

Finally, if day $t$ is the last day, we adopt the one-time mechanism to pay for day $t$'s data as well.

---

**Mechanism 2:** Multi-time data collecting mechanism.

---

Given a differentiable convex function $f$ with $f' \in [0, U]$ and $f^*$ bounded on $[0, U]$
**for** $t = 1, \ldots, T$ **do**

(1) On day $t$, ask all data providers to report their datasets $\widetilde{D}_1^{(t)}, \ldots, \widetilde{D}_n^{(t)}$.
(2) If $t$ is the last day $t = T$, use the payment rule of Mechanism 1 to pay for day $T$'s data or just give each data provider $B^{(T)}/n$.
(3) If $t > 1$, give the payments for day $t-1$ as follows. First compute all the scores $s_i$ as in (5). Then normalize the scores so that the total payment is no more than $B^{(t-1)}$. Let the range of the scores be $[L, R]$. Assign payments
$$r_i(\widetilde{D}_1^{(t-1)}, \ldots, \widetilde{D}_n^{(t-1)}) = \frac{B^{(t-1)}}{n} \cdot \frac{s_i - L}{R - L}.$$

**end for**

---

Our first result is that Mechanism 2 guarantees all the basic properties of a desirable mechanism.

**Theorem 6.1.** *Given any differentiable convex function $f$ that has (1) a bounded derivative $f' \in [0, U]$ and (2) the convex conjugate $f^*$ bounded on $[0, U]$, Mechanism 2 is IR, budget feasible, truthful and symmetric in all $T$ rounds.*

If we choose computable $f'$ and $f^*$ (e.g. $f'$ equal to the Logistic function), the payments will also be computable for finite-size $\Theta$ and for models in exponential family (Lemma 4.1). If we use a strictly convex function $f$ with $f' > 0$, then Mechanism 2 has basically the same sensitivity guarantee as Mechanism 1 in the first $T - 1$ rounds. We defer the sensitivity analysis to Appendix D.1. The missing proofs in this section can be found in Appendix D.2.

## 7  Discussion

Our work leaves some immediate open questions. Our one-time data acquisition mechanism requires a lower bound and an upper bound of PMI, which may be difficult to find for some models in the exponential family. Can we find a mechanism that would work for any data distribution, just as our multi-time data acquisition mechanism? Another interesting direction is to design stronger mechanisms to strengthen the sensitivity guarantees. Finally, our method incentivizes truthful reporting, but it is not guaranteed that datasets that give more accurate posteriors will receive higher payments (in expectation). It would be desirable if the mechanism could have this property as well.

An observation of our mechanism also raises an important issue of adversarial attacks for data acquisition. Our mechanism guarantees that the data holders who have some data will truthfully report the data at the equilibrium. But an adversary without any real data can submit some fake data and get a positive payoff. The situation is even worse if the adversary can create multiple fake accounts to submit data. This is tied to the individual rationality guarantee that we placed on our mechanism, which requires the payments to always be non-negative and which is generally considered an important property for incentivizing desirable behavior of strategic agents. However, this observation suggests that future work needs to carefully explore the tradeoff between guarantees for strategic agents and guarantees for adversarial agents for data acquisition problems.

## Broader Impact

The results in this work are mainly theoretical. They contribute to the ongoing efforts on encouraging data sharing, ensuring data quality and distributing values of generated from data back to data contributors.

## Acknowledgments and Disclosure of Funding

This work is supported by the National Science Foundation under grants CCF-1718549 and IIS-2007887.

## Footnotes

[1]This means that a data provider can report a different dataset without changing his reward as long as the dataset leads to the same prediction for the underlying parameters as his true dataset.

[2]A constant payment rule is just a trivial truthful payment rule.

[3]Using a fixed test set may encourage misreporting.

[4]This is not to say that the providers will update their prior for $\boldsymbol{\theta}^{(t)}$ using the data on first $t-1$ days. Because we assume that $\boldsymbol{\theta}^{(t)}$ is independent from $\boldsymbol{\theta}^{(t-1)}, \ldots, \boldsymbol{\theta}^{(t-1)}$, so the data on first $t-1$ days contains no information about $\boldsymbol{\theta}^{(t)}$. We use the same prior $p(\boldsymbol{\theta})$ throughout all $T$ days. What it means is that when the analyst decides the payment for day $t$ not only based on the report on day $t$ but also the historical reports, the providers may also use different strategies for different historical reports.

[5]WLOG, we can assume that $L < R$ here. Because $L = R$ implies that all agents' datasets are independent.

[6]The term $PMI(\cdot)$ in the payment function of [16] should actually be $1/PMI(\cdot)$. This is because when [16] cited Lemma 1 in [22], $q(\cdot)/p(\cdot)$ is mistakenly replaced by $p(\cdot)/q(\cdot)$.

[7]Here we assume that $PMI(\cdot)$ is non-zero. For $PMI(\cdot) = 0$, we can just do the same as in the one-time acquisition mechanism.

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
