[Supplementary Material]

# A    Mathematical Background

Our mechanisms are built with some important mathematical tools. First, in probability theory, an $f$-divergence is a function that measures the difference between two probability distributions.

**Definition A.1** ($f$-divergence)**.** *Given a convex function $f$ with $f(1) = 0$, for two distributions over $\Omega$, $p, q \in \Delta\Omega$, define the $f$-divergence of $p$ and $q$ to be*

$$D_f(p, q) = \int_{\omega \in \Omega} p(\omega) f\left(\frac{q(\omega)}{p(\omega)}\right).$$

In duality theory, the convex conjugate of a function is defined as follows.

**Definition A.2** (Convex conjugate)**.** *For any function $f : \mathbb{R} \to \mathbb{R}$, define the convex conjugate function of $f$ as*

$$f^*(y) = \sup_x xy - f(x).$$

Then the following inequality ([22, 16]) holds.

**Lemma A.1** (Lemma 1 in [22])**.** *For any differentiable convex function $f$ with $f(1) = 0$, any two distributions over $\Omega$, $p, q \in \Delta\Omega$, let $\mathcal{G}$ be the set of all functions from $\Omega$ to $\mathbb{R}$, then we have*

$$D_f(p, q) \geq \sup_{g \in \mathcal{G}} \int_{\omega \in \Omega} g(\omega) q(\omega) - f^*(g(\omega)) p(\omega) \, d\omega = \sup_{g \in \mathcal{G}} \mathbb{E}_q g - \mathbb{E}_p f^*(g).$$

*A function $g$ achieves equality if and only if $g(\omega) \in \partial f\left(\frac{q(\omega)}{p(\omega)}\right) \forall \omega$ with $p(\omega) > 0$, where $\partial f\left(\frac{q(\omega)}{p(\omega)}\right)$ represents the subdifferential of $f$ at point $q(\omega)/p(\omega)$.*

The $f$-mutual information of two random variables is a measure of the mutual dependence of two random variables, which is defined as the $f$-divergence between their joint distribution and the product of their marginal distributions.

**Definition A.3** (Kronecker product)**.** *Consider two matrices $A \in \mathbb{R}^{m \times n}$ and $B \in \mathbb{R}^{p \times q}$. The Kronecker product of $A$ and $b$, denoted as $A \otimes B$, is defined as the following $pm \times qn$ matrix:*

$$A \otimes B = \begin{bmatrix} a_{11}B & \cdots & a_{1n}B \\ \vdots & \ddots & \vdots \\ a_{m1}B & \cdots & a_{mn}B \end{bmatrix}.$$

**Definition A.4** ($f$-mutual information and pointwise MI)**.** *Let $(X, Y)$ be a pair of random variables with values over the space $\mathcal{X} \times \mathcal{Y}$. If their joint distribution is $p_{X,Y}$ and marginal distributions are $p_X$ and $p_Y$, then given a convex function $f$ with $f(1) = 0$, the $f$-mutual information between $X$ and $Y$ is*

$$I_f(X; Y) = D_f(p_{X,Y}, p_X \otimes p_Y) = \int_{x \in \mathcal{X}, y \in \mathcal{Y}} p_{X,Y}(x, y) f\left(\frac{p_X(x) \cdot p_Y(y)}{p_{X,Y}(x, y)}\right).$$

*We define function $K(x, y)$ as the reciprocal of the ratio inside $f$,*

$$K(x, y) = \frac{p_{X,Y}(x, y)}{p_X(x) \cdot p_Y(y)}.$$

If two random variables are independent conditioning on another random variable, we have the following formula for the function $K$.

**Lemma A.2.** *When random variables $X, Y$ are independent conditioning on $\boldsymbol{\theta}$, for any pair of $(x, y) \in \mathcal{X} \times \mathcal{Y}$, we have*

$$K(x, y) = \sum_{\boldsymbol{\theta} \in \Theta} \frac{p(\boldsymbol{\theta}|x) p(\boldsymbol{\theta}|y)}{p(\boldsymbol{\theta})}$$

*if $|\Theta|$ is finite, and*

$$K(x, y) = \int_{\boldsymbol{\theta} \in \Theta} \frac{p(\boldsymbol{\theta}|x) p(\boldsymbol{\theta}|y)}{p(\boldsymbol{\theta})} \, d\boldsymbol{\theta}$$

*if $\Theta \subseteq \mathbb{R}^m$.*

*Proof.* We only prove the second equation for $\Theta \subseteq \mathbb{R}^m$ as the proof for finite $\Theta$ is totally similar.

$$K(x, y) = \frac{p(x, y)}{p(x) \cdot p(y)}$$

$$= \frac{\int_{\boldsymbol{\theta} \in \Theta} p(x|\boldsymbol{\theta})p(y|\boldsymbol{\theta})p(\boldsymbol{\theta}) \, d\boldsymbol{\theta}}{p(x) \cdot p(y)}$$

$$= \int_{\boldsymbol{\theta} \in \Theta} \frac{p(\boldsymbol{\theta}|x)p(\boldsymbol{\theta}|y)}{p(\boldsymbol{\theta})} \, d\boldsymbol{\theta},$$

where the last equation uses Bayes' Law. $\qquad \square$

**Definition A.5** (Exponential family [21])**.** *A probability density function or probability mass function* $p(\mathbf{x}|\boldsymbol{\theta})$*, for* $\mathbf{x} = (x_1, \ldots, x_n) \in \mathcal{X}^n$ *and* $\boldsymbol{\theta} \in \Theta \subseteq \mathbb{R}^m$ *is said to be in the* exponential family *in canonical form if it is of the form*

$$p(\mathbf{x}|\boldsymbol{\theta}) = h(\mathbf{x}) \exp\left[\boldsymbol{\theta}^T \phi(\mathbf{x}) - A(\boldsymbol{\theta})\right] \tag{6}$$

*where* $A(\boldsymbol{\theta}) = \log \int_{\mathcal{X}^m} h(\mathbf{x}) \exp\left[\boldsymbol{\theta}^T \phi(\mathbf{x})\right]$*. The conjugate prior with parameters* $\nu_0, \overline{\boldsymbol{\tau}}_0$ *for* $\boldsymbol{\theta}$ *has the form*

$$p(\boldsymbol{\theta}) = \mathcal{P}(\boldsymbol{\theta}|\nu_0, \overline{\boldsymbol{\tau}}_0) = g(\nu_0, \overline{\boldsymbol{\tau}}_0) \exp\left[\nu_0 \boldsymbol{\theta}^T \overline{\boldsymbol{\tau}}_0 - \nu_0 A(\boldsymbol{\theta})\right]. \tag{7}$$

*Let* $\overline{s} = \frac{1}{n} \sum_{i=1}^n \phi(x_i)$*. Then the posterior of* $\boldsymbol{\theta}$ *is of the form*

$$p(\boldsymbol{\theta}|\mathbf{x}) \propto \exp\left[\boldsymbol{\theta}^T (\nu_0 \overline{\boldsymbol{\tau}}_0 + n\overline{s}) - (\nu_0 + n) A(\boldsymbol{\theta})\right]$$

$$= \mathcal{P}\left(\boldsymbol{\theta}|\nu_0 + n, \frac{\nu_0 \overline{\boldsymbol{\tau}}_0 + n\overline{s}}{\nu_0 + n}\right),$$

*where* $\mathcal{P}\left(\boldsymbol{\theta}|\nu_0 + n, \frac{\nu_0 \overline{\boldsymbol{\tau}}_0 + n\overline{s}}{\nu_0 + n}\right)$ *is the conjugate prior with parameters* $\nu_0 + n$ *and* $\frac{\nu_0 \overline{\boldsymbol{\tau}}_0 + n\overline{s}}{\nu_0 + n}$*.*

**Lemma A.3.** *Let* $\boldsymbol{\theta}$ *be the parameters of a pdf in the exponential family. Let* $\mathcal{P}(\boldsymbol{\theta}|\nu, \overline{\boldsymbol{\tau}}) = g(\nu, \overline{\boldsymbol{\tau}}) \exp\left[\nu \boldsymbol{\theta}^T \overline{\boldsymbol{\tau}} - \nu A(\boldsymbol{\theta})\right]$ *denote the conjugate prior for* $\boldsymbol{\theta}$ *with parameters* $\nu, \overline{\boldsymbol{\tau}}$*. For any three distributions of* $\boldsymbol{\theta}$*,*

$$p_1(\boldsymbol{\theta}) = \mathcal{P}(\boldsymbol{\theta}|\nu_1, \overline{\boldsymbol{\tau}}_1),$$
$$p_2(\boldsymbol{\theta}) = \mathcal{P}(\boldsymbol{\theta}|\nu_2, \overline{\boldsymbol{\tau}}_2),$$
$$p_0(\boldsymbol{\theta}) = \mathcal{P}(\boldsymbol{\theta}|\nu_0, \overline{\boldsymbol{\tau}}_0),$$

*we have*

$$\int_{\boldsymbol{\theta} \in \Theta} \frac{p_1(\boldsymbol{\theta})p_2(\boldsymbol{\theta})}{p_0(\boldsymbol{\theta})} \, d\boldsymbol{\theta} = \frac{g(\nu_1, \overline{\boldsymbol{\tau}}_1)g(\nu_2, \overline{\boldsymbol{\tau}}_2)}{g(\nu_0, \overline{\boldsymbol{\tau}}_0)g(\nu_1 + \nu_2 - \nu_0, \frac{\nu_1 \overline{\boldsymbol{\tau}}_1 + \nu_2 \overline{\boldsymbol{\tau}}_2 - \nu_0 \overline{\boldsymbol{\tau}}_0}{\nu_1 + \nu_2 - \nu_0})}.$$

*Proof.* To compute the integral, we first write $p_1(\boldsymbol{\theta})$, $p_2(\boldsymbol{\theta})$ and $p_3(\boldsymbol{\theta})$ in full,

$$p_1(\boldsymbol{\theta}) = \mathcal{P}(\boldsymbol{\theta}|\nu_1, \overline{\boldsymbol{\tau}}_1) = g(\nu_1, \overline{\boldsymbol{\tau}}_1) \exp\left[\nu_1 \boldsymbol{\theta}^T \overline{\boldsymbol{\tau}}_1 - \nu_1 A(\boldsymbol{\theta})\right],$$
$$p_2(\boldsymbol{\theta}) = \mathcal{P}(\boldsymbol{\theta}|\nu_2, \overline{\boldsymbol{\tau}}_2) = g(\nu_2, \overline{\boldsymbol{\tau}}_2) \exp\left[\nu_2 \boldsymbol{\theta}^T \overline{\boldsymbol{\tau}}_2 - \nu_2 A(\boldsymbol{\theta})\right],$$
$$p_0(\boldsymbol{\theta}) = \mathcal{P}(\boldsymbol{\theta}|\nu_0, \overline{\boldsymbol{\tau}}_0) = g(\nu_0, \overline{\boldsymbol{\tau}}_0) \exp\left[\nu_0 \boldsymbol{\theta}^T \overline{\boldsymbol{\tau}}_0 - \nu_0 A(\boldsymbol{\theta})\right].$$

Then we have the integral equal to

$$\int_{\boldsymbol{\theta} \in \Theta} \frac{p_1(\boldsymbol{\theta})p_2(\boldsymbol{\theta})}{p_0(\boldsymbol{\theta})} \, d\boldsymbol{\theta}$$

$$= \int_{\boldsymbol{\theta} \in \Theta} \frac{g(\nu_1, \overline{\boldsymbol{\tau}}_1) \exp\left[\nu_1 \boldsymbol{\theta}^T \overline{\boldsymbol{\tau}}_1 - \nu_1 A(\boldsymbol{\theta})\right] g(\nu_2, \overline{\boldsymbol{\tau}}_2) \exp\left[\nu_2 \boldsymbol{\theta}^T \overline{\boldsymbol{\tau}}_2 - \nu_2 A(\boldsymbol{\theta})\right]}{g(\nu_0, \overline{\boldsymbol{\tau}}_0) \exp\left[\nu_0 \boldsymbol{\theta}^T \overline{\boldsymbol{\tau}}_0 - \nu_0 A(\boldsymbol{\theta})\right]} \, d\boldsymbol{\theta}$$

$$= \frac{g(\nu_1, \overline{\boldsymbol{\tau}}_1)g(\nu_2, \overline{\boldsymbol{\tau}}_2)}{g(\nu_0, \overline{\boldsymbol{\tau}}_0)} \int_{\boldsymbol{\theta} \in \Theta} \exp\left[\boldsymbol{\theta}^T (\nu_1 \overline{\boldsymbol{\tau}}_1 + \nu_2 \overline{\boldsymbol{\tau}}_2 - \nu_0 \overline{\boldsymbol{\tau}}_0) - A(\boldsymbol{\theta})(\nu_1 + \nu_2 - \nu_0)\right] \, d\boldsymbol{\theta}$$

$$= \frac{g(\nu_1, \overline{\boldsymbol{\tau}}_1)g(\nu_2, \overline{\boldsymbol{\tau}}_2)}{g(\nu_0, \overline{\boldsymbol{\tau}}_0)} \cdot \frac{1}{g(\nu_1 + \nu_2 - \nu_0, \frac{\nu_1 \overline{\boldsymbol{\tau}}_1 + \nu_2 \overline{\boldsymbol{\tau}}_2 - \nu_0 \overline{\boldsymbol{\tau}}_0}{\nu_1 + \nu_2 - \nu_0})}.$$

The last equality is because

$$g\left(\nu_1 + \nu_2 - \nu_0, \frac{\nu_1\overline{\boldsymbol{\tau}}_1 + \nu_2\overline{\boldsymbol{\tau}}_2 - \nu_0\overline{\boldsymbol{\tau}}_0}{\nu_1 + \nu_2 - \nu_0}\right) \exp\left[\boldsymbol{\theta}^T(\nu_1\overline{\boldsymbol{\tau}}_1 + \nu_2\overline{\boldsymbol{\tau}}_2 - \nu_0\overline{\boldsymbol{\tau}}_0) - A(\boldsymbol{\theta})(\nu_1 + \nu_2 - \nu_0)\right]$$

is the pdf

$$p\left(\boldsymbol{\theta}|\nu_1 + \nu_2 - \nu_0, \frac{\nu_1\overline{\boldsymbol{\tau}}_1 + \nu_2\overline{\boldsymbol{\tau}}_2 - \nu_0\overline{\boldsymbol{\tau}}_0}{\nu_1 + \nu_2 - \nu_0}\right)$$

and thus has the integral over $\boldsymbol{\theta}$ equal to 1. $\qquad\square$

## B  Missing proof for Lemma 3.1

**Lemma B.1** (Lemma 3.1). *When $D_1, \ldots, D_n$ are independent conditioned on $\boldsymbol{\theta}$, for any $(D_1, \ldots, D_n)$ and $(\widetilde{D}_1, \ldots, \widetilde{D}_n)$, if $p(\boldsymbol{\theta}|D_i) = p(\boldsymbol{\theta}|\widetilde{D}_i) \ \forall i$, then $p(\boldsymbol{\theta}|D_1, \ldots, D_n) = p(\boldsymbol{\theta}|\widetilde{D}_1, \ldots, \widetilde{D}_n)$.*

*Proof.* Suppose $\forall i, p(\boldsymbol{\theta}|D_i) = p(\boldsymbol{\theta}|D_i')$, then we have

$$
\begin{aligned}
p(\boldsymbol{\theta}|D_1, D_2, \cdots, D_n) &= \frac{p(D_1, D_2, \cdots, D_n, \boldsymbol{\theta})}{p(D_1, D_2, \cdots, D_n)} \\
&= \frac{p(D_1, D_2, \cdots, D_n|\boldsymbol{\theta}) \cdot p(\boldsymbol{\theta})}{p(D_1, D_2, \cdots, D_n)} \\
&= \frac{p(D_1|\boldsymbol{\theta}) \cdot p(D_2|\boldsymbol{\theta}) \cdots p(D_n|\boldsymbol{\theta}) \cdot p(\boldsymbol{\theta})}{p(D_1, D_2, \cdots, D_n)} \\
&= \frac{p(D_1, \boldsymbol{\theta}) \cdot p(D_2, \boldsymbol{\theta}) \cdots p(D_n, \boldsymbol{\theta}) \cdot p(\boldsymbol{\theta})}{p(D_1, D_2, \cdots, D_n) \cdot p^n(\boldsymbol{\theta})} \\
&= \frac{p(\boldsymbol{\theta}|D_1) \cdot p(\boldsymbol{\theta}|D_2) \cdots p(\boldsymbol{\theta}|D_n) \cdot p(D_1) \cdot p(D_2) \cdots p(D_n)}{p(D_1, D_2, \cdots, D_n) \cdot p^{n-1}(\boldsymbol{\theta})} \\
&\propto \frac{p(\boldsymbol{\theta}|D_1) \cdot p(\boldsymbol{\theta}|D_2) \cdots p(\boldsymbol{\theta}|D_n)}{p^{n-1}(\boldsymbol{\theta})}.
\end{aligned}
$$

Similarly, we have

$$p(\boldsymbol{\theta}|D_1', D_2', \cdots, D_n') \propto \frac{p(\boldsymbol{\theta}|D_1') \cdot p(\boldsymbol{\theta}|D_2') \cdots p(\boldsymbol{\theta}|D_n')}{p^{n-1}(\boldsymbol{\theta})},$$

since the analyst calculate the posterior by normalize the terms, we have

$$p(\boldsymbol{\theta}|D_1, D_2, \cdots, D_n) = p(\boldsymbol{\theta}|D_1', D_2', \cdots, D_n').$$

$\qquad\square$

## C  One-time data acquisition

### C.1  An example of applying peer prediction

The mechanism is as follows.

---
**Mechanism 3:** One-time data collecting mechanism by using Brier Score.

---
(1) Ask all data providers to report their datasets $\widetilde{D}_1, \ldots, \widetilde{D}_n$.
(2) For all $D_{-i}$, calculate probability $p(D_{-i}|D_i)$ by the reported $D_i$ and $p(D_i|\boldsymbol{\theta})$.
(3) The Brier score for agent $i$ is $s_i = 1 - \frac{1}{|D_{-i}|}\sum_{D_{-i}}(p(D_{-i}|\widetilde{D}_i) - \mathbb{I}[D_{-i} = \widetilde{D}_{-i}])^2$,
where $\mathbb{I}[D_{-i} = \widetilde{D}_{-i}] = 1$ if $D_{-i}$ is the same as the reported $\widetilde{D}_{-i}$ and 0 otherwise.
(4) The final payment for agent $i$ is $r_i = \frac{B \cdot s_i}{n}$.

---

This payment function is actually the mean square error of the reported distribution on $D_{-i}$. It is based on the Brier score which is first proposed in [3] and is a well-known bounded proper scoring rule. The payments of the mechanism are always bounded between 0 and 1.

**Theorem C.1.** *Mechanism 3 is IR, truthful, budget feasible, symmetric.*

*Proof.* The symmetric property is easy to verify. Moreover, since the payment for each agent is in the interval $[0, 1]$, the mechanism is then budget feasible and IR. We only need to prove the truthfulness. Suppose that all the other agents except $i$ reports truthfully. Agent $i$ has true dataset $D_i$ and reports $\widetilde{D}_i$. Since in the setting, the analyst is able to calculate $p(D_{-i}|D_i)$, then if the agent receives $s_i$ as their payment, from agent $i$'s perspective, his expected revenue is then:

$$Rev'_i = \sum_{D_{-i}} p(D_{-i}|D_i) \cdot \left( 1 - \sum_{D'_{-i}} (p(D'_{-i}|\widetilde{D}_i) - \mathbb{I}[D'_{-i} = D_{-i}])^2 \right)$$

$$= -\sum_{D_{-i}} p(D_{-i}|D_i) \left( \sum_{D'_{-i}} \left( p(D'_{-i}|\widetilde{D}_i)^2 \right) - 2p(D_{-i}|\widetilde{D}_i) \right)$$

$$= \sum_{D_{-i}} \left( -p(D_{-i}|\widetilde{D}_i)^2 + 2p(D_{-i}|\widetilde{D}_i)p(D_{-i}|D_i) \right)$$

Since the function $-x^2 + 2ax$ is maximized when $x = a$, the revenue $Rev'_i$ is maximized when $\forall D_{-i}, p(D_{-i}|D_{-i}) = p(D_{-i}|D_i)$. Since the real payment $r_i$ is a linear transformation of $s_i$ and the coefficients are independent of the reported datasets, reporting the dataset with the true posterior will still maximize the agent's revenue and the mechanism is truthful. $\square$

## C.2 Bounding log-PMI: discrete case

In this section, we give a method to compute the bounds of the log-PMI score when $|\Theta|$ is finite. First we give the upper bound of the PMI. We have for any $i, D_i \in \mathbb{D}_i(D_{-i})$

$$PMI(D_i, D_{-i}) \leq \max_{i, D'_{-i}, D'_i \in \mathbb{D}_i(D'_{-i})} \{PMI(D'_i, D'_{-i})\}$$

$$= \max_{i, D'_{-i}, D'_i \in \mathbb{D}_i(D'_{-i})} \left\{ \sum_{\boldsymbol{\theta} \in \Theta} \frac{p(\boldsymbol{\theta}|D'_i)p(\boldsymbol{\theta}|D'_{-i})}{p(\boldsymbol{\theta})} \right\}$$

$$\leq \max_{i, D'_i} \left\{ \sum_{\boldsymbol{\theta} \in \Theta} \frac{p(\boldsymbol{\theta}|D'_i)}{\min_{\boldsymbol{\theta}}\{p(\boldsymbol{\theta})\}} \right\}$$

$$\leq \frac{1}{\min_{\boldsymbol{\theta}}\{p(\boldsymbol{\theta})\}}.$$

The last inequality is because we have $\sum_{\boldsymbol{\theta}} p(\boldsymbol{\theta}|D'_i) = 1$.

Since we have assumed that $p(\boldsymbol{\theta})$ is positive, the term $\frac{1}{\min_{\boldsymbol{\theta}}\{p(\boldsymbol{\theta})\}}$ could then be computed and is finite. Thus we just let $R$ be $\log\left(\frac{1}{\min_{\boldsymbol{\theta}}\{p(\boldsymbol{\theta})\}}\right)$. Then we need to calculate a lower bound of the score. We have for any $i, D_{-i}$ and $D_i \in \mathbb{D}_i(D_{-i})$

$$PMI(D_i, D_{-i}) = \sum_{\boldsymbol{\theta} \in \Theta} \frac{p(\boldsymbol{\theta}|D_i)p(\boldsymbol{\theta}|D_{-i})}{p(\boldsymbol{\theta})} \geq \sum_{\boldsymbol{\theta} \in \Theta} p(\boldsymbol{\theta}|D_i)p(\boldsymbol{\theta}|D_{-i}). \tag{8}$$

**Claim C.1.** *Let $D = \{d^{(1)}, \ldots, d^{(N)}\}$ be a dataset with $N$ data points that are i.i.d. conditioning on $\boldsymbol{\theta}$. Let $\mathcal{D}$ be the support of the data points $d$. Define*

$$T = \frac{\max_{\boldsymbol{\theta} \in \Theta} p(\boldsymbol{\theta})}{\min_{\boldsymbol{\theta} \in \Theta} p(\boldsymbol{\theta})}, \quad U(\mathcal{D}) = \max_{\boldsymbol{\theta} \in \Theta, d \in \mathcal{D}} p(\boldsymbol{\theta}|d) \Big/ \min_{\boldsymbol{\theta} \in \Theta, d \in \mathcal{D}: p(\boldsymbol{\theta}|d)>0} p(\boldsymbol{\theta}|d),$$

*Then we have*

$$\frac{\max_{\boldsymbol{\theta} \in \Theta} p(\boldsymbol{\theta}|D)}{\min_{\boldsymbol{\theta}: p(\boldsymbol{\theta}|D)>0} p(\boldsymbol{\theta}|D)} \leq U(\mathcal{D})^N \cdot T^{N-1}.$$

*Proof.* By Lemma 3.1, we have

$$p(\boldsymbol{\theta}|D) \propto \frac{\prod_j p(\boldsymbol{\theta}|d^{(j)})}{p(\boldsymbol{\theta})^{N-1}},$$

for a fixed $D$, it must hold that

$$\frac{\max_{\boldsymbol{\theta} \in \Theta} p(\boldsymbol{\theta}|D)}{\min_{\boldsymbol{\theta}:p(\boldsymbol{\theta}|D)>0} p(\boldsymbol{\theta}|D)} \leq U(\mathcal{D})^N \cdot T^{N-1}.$$

$\square$

**Claim C.2.** *For any two datasets $D_i$ and $D_j$ with $N_i$ and $N_j$ data points respectively, let $\mathcal{D}_i$ be the support of the data points in $D_i$ and let $\mathcal{D}_j$ be the support of the data points in $D_j$. Then*

$$\frac{\max_{\boldsymbol{\theta} \in \Theta} p(\boldsymbol{\theta}|D_i, D_j)}{\min_{\boldsymbol{\theta}:p(\boldsymbol{\theta}|D_i,D_j)>0} p(\boldsymbol{\theta}|D_i, D_j)} \leq U(\mathcal{D}_i)^{N_i} \cdot U(\mathcal{D}_j)^{N_j} \cdot T^{N_i+N_j-1}.$$

*Proof.* Again by Lemma 3.1, we have

$$p(\boldsymbol{\theta}|D_i, D_j) \propto \frac{p(\boldsymbol{\theta}|D_i)p(\boldsymbol{\theta}|D_j)}{p(\boldsymbol{\theta})}.$$

Combine it with Claim C.1, we prove the statement. $\square$

Then for any $D_i$, since $\sum_{\boldsymbol{\theta} \in \Theta} p(\boldsymbol{\theta}|D_i) = 1$, by Claim C.1,

$$\min_{\boldsymbol{\theta}:p(\boldsymbol{\theta}|D_i)>0} p(\boldsymbol{\theta}|D_i) \geq \frac{1}{1 + |\Theta| \cdot U(\mathcal{D}_i)^{N_i} \cdot T^{N_i-1}} \triangleq \eta(\mathcal{D}_i, N_i).$$

And for any $D_{-i}$, since $\sum_{\boldsymbol{\theta} \in \Theta} p(\boldsymbol{\theta}|D_{-i}) = 1$, by Claim C.2,

$$\min_{\boldsymbol{\theta}:p(\boldsymbol{\theta}|D_{-i})>0} p(\boldsymbol{\theta}|D_{-i}) \geq \frac{1}{1 + |\Theta| \cdot \Pi_{j \neq i} U(\mathcal{D}_j)^{N_j} \cdot T^{\sum_{j \neq i} N_j - 1}} \triangleq \eta(\mathcal{D}_{-i}, N_{-i}).$$

Finally, for any $i, D_{-i}$, and $D_i \in \mathbb{D}_i(D_{-i})$, according to (8),

$$PMI(D_i, D_{-i}) \geq \sum_{\boldsymbol{\theta} \in \Theta} p(\boldsymbol{\theta}|D_i)p(\boldsymbol{\theta}|D_{-i}) \geq \eta(\mathcal{D}_i, N_i) \cdot \eta(\mathcal{D}_{-i}, N_{-i}).$$

The last inequality is because $D_i \in \mathbb{D}_i(D_{-i})$ and there must exists $\boldsymbol{\theta} \in \Theta$ so that both $p(\boldsymbol{\theta}|D_i)$ and $p(\boldsymbol{\theta}|D_{-i})$ are non-zero. Both $\eta(\mathcal{D}_i, N_i)$ and $\eta(\mathcal{D}_{-i}, N_{-i})$ can be computed in polynomial time. Take minimum over $i$, we find the lower bound for PMI.

### C.3 Bounding log-PMI: continuous case

Consider estimating the mean $\mu$ of a univariate Gaussian $\mathcal{N}(x|\mu, \sigma^2)$ with known variance $\sigma^2$. Let $D = \{x_1, \ldots, x_N\}$ be the dataset and denote the mean by $\bar{x} = \frac{1}{N} \sum_j x_j$. We use the Gaussian conjugate prior,

$$\mu \sim \mathcal{N}(\mu|\mu_0, \sigma_0^2).$$

Then according to [20], the posterior of $\mu$ is equal to

$$p(\mu|D) = \mathcal{N}(\mu|\mu_N, \sigma_N^2),$$

where

$$\frac{1}{\sigma_N^2} = \frac{1}{\sigma_0^2} + \frac{N}{\sigma^2}$$

only depends on the number of data points.

By Lemma 4.1, we know that the payment function for exponential family is in the form of

$$PMI(D_i, D_{-i}) = \frac{g(\nu_i, \overline{\boldsymbol{\tau}}_i)g(\nu_{-i}, \overline{\boldsymbol{\tau}}_{-i})}{g(\nu_0, \overline{\boldsymbol{\tau}}_0)g(\nu_i + \nu_{-i} - \nu_0, \frac{\nu_i \overline{\boldsymbol{\tau}}_i + \nu_{-i} \overline{\boldsymbol{\tau}}_{-i} - \nu_0 \overline{\boldsymbol{\tau}}_0}{\nu_i + \nu_{-i} - \nu_0})}.$$

The normalization term for Gaussian is $\frac{1}{\sqrt{2\pi\sigma^2}}$, so we have

$$PMI(D_i, D_{-i}) = \frac{\sqrt{\frac{1}{\sigma_0^2} + \frac{N_i}{\sigma^2}}\sqrt{\frac{1}{\sigma_0^2} + \frac{N_{-i}}{\sigma^2}}}{\sqrt{\frac{1}{\sigma_0^2}}\sqrt{\frac{1}{\sigma_0^2} + \frac{N_i + N_{-i}}{\sigma^2}}}.$$

When the total number of data points has an upper bound $N_{\max}$, each of the square root term should be bounded in the interval

$$\left[\frac{1}{\sigma_0}, \sqrt{\frac{1}{\sigma_0^2} + \frac{N_{max}}{\sigma^2}}\right]$$

Therefore $PMI(D_i, D_{-i})$ is bounded in the interval

$$\left[\left(1 + N_{max}\sigma_0^2/\sigma^2\right)^{-1/2}, 1 + N_{max}\sigma_0^2/\sigma^2\right].$$

## C.4  Sensitivity analysis for the exponential family

If we are estimating the mean $\mu$ of a univariate Gaussian $\mathcal{N}(x|\mu, \sigma^2)$ with known variance $\sigma^2$. Let $D = \{x_1, \ldots, x_N\}$ be the dataset and denote the mean by $\bar{x} = \frac{1}{N}\sum_j x_j$. We use the Gaussian conjugate prior,

$$\mu \sim \mathcal{N}(\mu|\mu_0, \sigma_0^2).$$

Then according to [20], the posterior of $\mu$ is equal to

$$p(\mu|D) = \mathcal{N}(\mu|\mu_N, \sigma_N^2),$$

where

$$\frac{1}{\sigma_N^2} = \frac{1}{\sigma_0^2} + \frac{N}{\sigma^2}$$

only depends on the number of data points. Since the normalization term $\frac{1}{\sqrt{2\pi\sigma^2}}$ of Gaussian distributions only depends on the variance, function $h(\cdot)$ defined in (12)

$$h_{D_{-i}}(N_i, \bar{x}_i) = \frac{g(\nu_i, \overline{\boldsymbol{\tau}}_i)}{g(\nu_i + \nu_{-i} - \nu_0, \frac{\nu_i\overline{\boldsymbol{\tau}}_i + \nu_{-i}\overline{\boldsymbol{\tau}}_{-i} - \nu_0\overline{\boldsymbol{\tau}}_0}{\nu_i + \nu_{-i} - \nu_0})}$$

$$= \sqrt{\frac{1}{\sigma_0^2} + \frac{N_i}{\sigma^2}} \Big/ \sqrt{\frac{1}{\sigma_0^2} + \frac{N_i + N_{-i}}{\sigma^2}}$$

will only be changed if the number of data points $N_i$ changes, which means that the mechanism will be sensitive to replication and withholding, but not necessarily other types of manipulations.

If we are estimating the mean $\mu$ of a Bernoulli distribution $Ber(x|\mu)$. Let $D = \{x_1, \ldots, x_N\}$ be the data points. Denote by $\alpha = \sum_i x_i$ the number of ones and denote by $\beta = \sum_i 1 - x_i$ the number of zeros. The conjugate prior is the Beta distribution,

$$p(\mu) = \text{Beta}(\mu|\alpha_0, \beta_0) = \frac{1}{B(\alpha_0, \beta_0)}\mu^{\alpha_0 - 1}(1 - \mu)^{\beta_0 - 1}.$$

where $B(\alpha_0, \beta_0)$ is the Beta function

$$B(\alpha_0, \beta_0) = \frac{(\alpha_0 + \beta_0 - 1)!}{(\alpha_0 - 1)!(\beta_0 - 1)!}.$$

The posterior of $\mu$ is equal to

$$p(\mu|D) = \text{Beta}(\mu|\alpha_0 + \alpha, \beta_0 + \beta).$$

Then we have

$$h_{D_{-i}}(\alpha, \beta) = \frac{B(\alpha_0 + \alpha_i + \alpha_{-i}, \beta_0 + \beta_i + \beta_{-i})}{B(\alpha_0 + \alpha_i, \beta_0 + \beta_i)}$$

$$= \frac{(\alpha_0 + \beta_0 + N_i + N_{-i} - 1)!(\alpha_0 + \alpha_i - 1)!(\beta_0 + \beta_i - 1)!}{(\alpha_0 + \alpha_i + \alpha_{-i} - 1)!(\beta_0 + \beta_i + \beta_{-i} - 1)!(\alpha_0 + \beta_0 + N_i - 1)!}.$$

Define $A_i = \alpha_0 + \alpha_i - 1$ and $B_i = \beta_0 + \beta_i - 1$, since $N_i = \alpha_i + \beta_i$ and $N_{-i} = \alpha_{-i} + \beta_{-i}$, we have

$$h_{D_{-i}}(\alpha, \beta) = h_{\alpha_{-i}, \beta_{-i}}(A_i, B_i) = \frac{A_i! B_i! (A_i + B_i + \alpha_{-i} + \beta_{-i} + 1)!}{(A_i + \alpha_{-i})!(B_i + \beta_{-i})!(A_i + B_i + 1)!}$$

Now we are going to prove that for any two different pairs $(A_i, B_i)$ and $(A_i', B_i')$, there should always exists a pair $(\alpha_{-i}', \beta_{-i}')$ selected from the four pairs: $(\alpha_{-i}, \beta_i), (\alpha_{-i} + 1, \beta_i), (\alpha_{-i}, \beta_i + 1), (\alpha_{-i} + 1, \beta_i + 1)$, such that $h_{\alpha_{-i}', \beta_{-i}'}(A_i, B_i) \neq h_{\alpha_{-i}', \beta_{-i}'}(A_i', B_i')$.

Suppose that this does not hold, then there should exist two pairs $(A_i, B_i)$ and $(A_i', B_i')$ such that for each $(\alpha_{-i}', \beta_{-i}')$ in the four pairs, $h_{\alpha_{-i}', \beta_{-i}'}(A_i, B_i) = h_{\alpha_{-i}', \beta_{-i}'}(A_i', B_i')$.

Then by the two cases when $(\alpha_{-i}', \beta_{-i}') = (\alpha_{-i}, \beta_{-i})$ and $(\alpha_{-i} + 1, \beta_{-i})$ we can derive that

$$\frac{h_{\alpha_{-i}+1, \beta_{-i}}(A_i, B_i)}{h_{\alpha_{-i}, \beta_{-i}}(A_i, B_i)} = \frac{h_{\alpha_{-i}+1, \beta_{-i}}(A_i', B_i')}{h_{\alpha_{-i}, \beta_{-i}}(A_i', B_i')}$$

$$\frac{A_i + B_i + \alpha_{-i} + 1 + \beta_{-i} + 1}{A_i + \alpha_{-i} + 1} = \frac{A_i' + B_i' + \alpha_{-i} + 1 + \beta_{-i} + 1}{A_i' + \alpha_{-i} + 1}$$

$$(A_i + B_i - A_i' - B_i')(\alpha_{-i} + 1) + (A_i' - A_i)(\alpha_{-i} + \beta_{-i} + 2) + A_i' B_i - A_i B_i' = 0$$

Replacing $\beta_{-i}$ with $\beta_{-i} + 1$, we could get

$$(A_i + B_i - A_i' - B_i')(\alpha_{-i} + 1) + (A_i' - A_i)(\alpha_{-i} + \beta_{-i} + 3) + A_i' B_i - A_i B_i' = 0$$

Subtracting the last equation from this, we get $A_i' - A_i = 0$. Symmetrically, when $(\alpha_{-i}', \beta_{-i}') = (\alpha_{-i}, \beta_{-i})$ and $(\alpha_{-i}, \beta_{-i} + 1)$ and replacing $\alpha_{-i}$ with $\alpha_{-i} + 1$, we have $B_i' - B_i = 0$ and thus $(A_i, B_i) = (A_i', B_i')$. This contradicts to the assumption that $(A_i, B_i) \neq (A_i', B_i')$. Therefore for any two different pairs of reported data in the Bernoulli setting, at least one in the four others' reported data $(\alpha_{-i}, \beta_i), (\alpha_{-i} + 1, \beta_i), (\alpha_{-i}, \beta_i + 1), (\alpha_{-i} + 1, \beta_i + 1)$ would make the agent strictly truthfully report his posterior.

## C.5  Missing proofs

### C.5.1  Proof for Theorem 5.1 and Theorem 5.2

**Theorem C.2** (Theorem 5.1). *Mechanism 1 is IR, truthful, budget feasible, symmetric.*

We suppose that the dataset space of agent $i$ is $\mathcal{D}_i$. We first give the definitions of several matrices. These matrices are essential for our proofs, but they are unknown to the data analyst. Since the dataset $D_i$ consists of $N_i$ i.i.d data points drawn from the data generating matrix $G_i$, we define prediction matrix $P_i$ of agent $i$ to be a matrix with $|\mathcal{D}_i| = |\mathcal{D}|^{N_i}$ rows and $|\Theta|$ columns. Each column corresponds to a $\boldsymbol{\theta} \in \Theta$ and each row corresponds to a possible dataset $D_i \in \mathcal{D}_i$. The matrix element on the column corresponding to $\boldsymbol{\theta}$ and the row corresponding to $D_i$ is $p(D_i|\boldsymbol{\theta})$. Intuitively, this matrix is the posterior of agent $i$'s dataset conditioned on the parameter $\boldsymbol{\theta}$.

Similarly, we define the out-prediction matrix $P_{-i}$ of agent $i$ to be a matrix with $\prod_{j \neq i} |\mathcal{D}_j|$ rows and $|Y|$ columns. Each column corresponds to a $\boldsymbol{\theta} \in \Theta$ and each row corresponds to a possible dataset $D_{-i} \in \mathcal{D}_{-i}$. The element corresponding to $D_{-i}$ and $\boldsymbol{\theta}$ is $p(D_{-i}|\boldsymbol{\theta})$. In the proof, we also give a lower bound on the sensitiveness coefficient $\alpha$ related to these out-prediction matrices.

**Theorem C.3** (Theorem 5.2). *Mechanism 1 is sensitive if either condition holds:*

1. $\forall i, Q_{-i}$ has rank $|\Theta|$.

2. $\forall i, \sum_{i' \neq i} (rank_k(G_{i'}) - 1) \cdot N_{i'} + 1 \geq |\Theta|$.

*Moreover, it is $e_i \cdot \frac{B}{n(R-L)}$-sensitive for agent $i$, where $e_i$ is the smallest singular value of matrix $P_{-i}$.*

*Proof.* First, it is easy to verify that the mechanism is budget feasible because $s_i$ is bounded between $L$ and $R$. Let agent $i$'s expected revenue of Mechanism 1 be $Rev_i$. Then we have

$$Rev_i = \frac{B}{n} \cdot \left( \frac{\sum_{D_{-i} \in \mathbb{D}_i(D_{-i})} p(D_{-i}|D_i) \cdot \log PMI(\widetilde{D}_i, D_{-i}) - L}{R - L} \right).$$

We consider another revenue $Rev'_i \triangleq \sum_{D_{-i}} p(D_{-i}|D_i) \cdot \log\left(\sum_{\boldsymbol{\theta}} \frac{p(\boldsymbol{\theta}|\widetilde{D}_i) \cdot p(\boldsymbol{\theta}|D_{-i})}{p(\boldsymbol{\theta})}\right)$ assuming that $0 \cdot \log 0 = 0$. Then we have

$$
\begin{aligned}
Rev'_i &= \sum_{D_{-i}} p(D_{-i}|D_i) \cdot \log\left(\sum_{\boldsymbol{\theta}} \frac{p(\boldsymbol{\theta}|\widetilde{D}_i) \cdot p(\boldsymbol{\theta}|D_{-i})}{p(\boldsymbol{\theta})}\right) \\
&= \sum_{D_{-i}, D_i \in \mathbb{D}_i(D_{-i})} p(D_{-i}|D_i) \cdot \log PMI(\widetilde{D}_i, D_{-i}) \\
&\quad + \sum_{D_{-i}, D_i \notin \mathbb{D}_i(D_{-i})} p(D_{-i}|D_i) \cdot \log PMI(\widetilde{D}_i, D_{-i}) \\
&= \sum_{D_{-i}, D_i \in \mathbb{D}_i(D_{-i})} p(D_{-i}|D_i) \cdot \log PMI(\widetilde{D}_i, D_{-i}) + \sum_{D_{-i}, D_i \notin \mathbb{D}_i(D_{-i})} 0 \cdot \log 0 \\
&= \sum_{D_{-i}, D_i \in \mathbb{D}_i(D_{-i})} p(D_{-i}|D_i) \cdot \log PMI(\widetilde{D}_i, D_{-i}) \\
&= Rev_i \cdot \frac{n}{B} \cdot (R - L) + L.
\end{aligned}
$$

$Rev'_i$ is a linear transformation of $Rev_i$. The coefficients $L$, $R$, $\frac{n}{B}$ do not depend on $\widetilde{D}_i$. The ratio $\frac{n}{B} \cdot (R - L)$ is larger than 0. Therefore, the optimal reported $\widetilde{D}_i$ for $Rev_i$ should be the same as that for $Rev'_i$. If the a payment rule with revenue $Rev'_i$ is $e_i$ - sensitive for agent $i$, then the Mechanism 1 would then be $e_i \cdot \frac{B}{n \cdot (R-L)}$ - sensitive. In the following part, we prove that real dataset $D_i$ would maximize the revenue $Rev'_i$ and the $Rev'_i$ is $e_i \cdot \frac{B}{|\mathcal{N}| \cdot (R-L)}$ - sensitive for all the agents. Thus in the following parts we prove the revenue $Rev'_i$ is $e_i$ - sensitive for agent $i$.

$$
\begin{aligned}
Rev'_i &= \sum_{D_{-i}} p(D_{-i}|D_i) \cdot \log\left(\sum_{\boldsymbol{\theta}} \frac{p(\boldsymbol{\theta}|\widetilde{D}_i) \cdot p(\boldsymbol{\theta}|D_{-i})}{p(\boldsymbol{\theta})}\right) \\
&= \sum_{D_{-i}} p(D_{-i}|D_i) \cdot \log\left(\sum_{\boldsymbol{\theta}} \frac{p(\boldsymbol{\theta}|\widetilde{D}_i) \cdot p(\boldsymbol{\theta}, D_{-i})}{p(\boldsymbol{\theta})}\right) - \sum_{D_{-i}} p(D_{-i}|D_i) \cdot \log\left(p(D_{-i})\right) \\
&= \sum_{D_{-i}} p(D_{-i}|D_i) \cdot \log\left(\sum_{\boldsymbol{\theta}} \frac{p(\boldsymbol{\theta}|\widetilde{D}_i) \cdot p(\boldsymbol{\theta}, D_{-i})}{p(\boldsymbol{\theta})}\right) - C.
\end{aligned}
$$

Since the term $\sum_{D_{-i}} p(D_{-i}|D_i) \cdot \log\left(p(D_{-i})\right)$ does not depend on $\widetilde{D}_i$, agent $i$ could only manipulate to modify the term $\sum_{D_{-i}} p(D_{-i}|D_i) \cdot \log\left(\sum_{\boldsymbol{\theta}} \frac{p(\boldsymbol{\theta}|\widetilde{D}_i) \cdot p(\boldsymbol{\theta}, D_{-i})}{p(\boldsymbol{\theta})}\right)$. Since we have

$$
\begin{aligned}
\sum_{D_{-i}, \boldsymbol{\theta}} \frac{p(\boldsymbol{\theta}|\widetilde{D}_i) \cdot p(\boldsymbol{\theta}, D_{-i})}{p(\boldsymbol{\theta})} &= \sum_{\boldsymbol{\theta}} \frac{1}{p(\boldsymbol{\theta})} \left(\sum_{D_{-i}} p(\boldsymbol{\theta}|\widetilde{D}_i) \cdot p(\boldsymbol{\theta}, D_{-i})\right) \\
&= \sum_{\boldsymbol{\theta}} \frac{1}{p(\boldsymbol{\theta})} \left(p(\boldsymbol{\theta}|\widetilde{D}_i) \cdot p(\boldsymbol{\theta})\right) \\
&= \sum_{\boldsymbol{\theta}} p(\boldsymbol{\theta}|\widetilde{D}_i) \\
&= 1,
\end{aligned}
$$

Since we have $\sum_{D_{-i}} \left(\sum_{\boldsymbol{\theta}} \frac{p(\boldsymbol{\theta}|\widetilde{D}_i) \cdot p(\boldsymbol{\theta}, D_{-i})}{p(\boldsymbol{\theta})}\right) = 1$, we could view the term $\sum_{\boldsymbol{\theta}} \frac{p(\boldsymbol{\theta}|\widetilde{D}_i) \cdot p(\boldsymbol{\theta}, D_{-i})}{p(\boldsymbol{\theta})}$ as a probability distribution on the variable $D_{-i}$. Since it depends on $\widetilde{D}_i$, we denote it as $\widetilde{p}(D_{-i}|\widetilde{D}_i)$. Since if we fix a distributions $p(\sigma)$, then the distribution $q(\sigma)$ that maximizes $\sum_{\sigma} p(\sigma) \log q(\sigma)$ should be the same as $p$. (If we assume that $0 \cdot \log 0 = 0$, this still holds.) When agent $i$ report

truthfully,

$$\sum_{\boldsymbol{\theta}} \frac{p(\boldsymbol{\theta}|D_i) \cdot p(\boldsymbol{\theta}, D_{-i})}{p(\boldsymbol{\theta})} = \sum_{\boldsymbol{\theta}} \frac{p(D_i, \boldsymbol{\theta}) \cdot p(D_{-i}, \boldsymbol{\theta})}{p(D_i) \cdot p(\boldsymbol{\theta})}$$

$$= \sum_{\boldsymbol{\theta}} \frac{p(D_i|\boldsymbol{\theta}) \cdot p(D_{-i}, \boldsymbol{\theta})}{p(D_i)}$$

$$= \sum_{\boldsymbol{\theta}} \frac{p(D_i|\boldsymbol{\theta}) \cdot p(D_{-i}|\boldsymbol{\theta}) \cdot p(\boldsymbol{\theta})}{p(D_i)}$$

$$= \sum_{\boldsymbol{\theta}} \frac{p(D_i, D_{-i}, \boldsymbol{\theta})}{p(D_i)}$$

$$= p(D_{-i}|D_i).$$

The data provider can always maximize $Rev_i'$ by truthfully reporting $D_i$. And we have proven the truthfulness of the mechanism.

Then we need to prove the relation between the sensitiveness of the mechanism and the out-prediction matrices. When Alice reports $\widetilde{D}_i$ the revenue difference from truthfully report is then

$$\Delta_{Rev_i'} = \sum_{D_{-i}} p(D_{-i}|D_i) \log p(D_{-i}|D_i) - \sum_{D_{-i}} p(D_{-i}|D_i) \log \widetilde{p}(D_{-i}|D_i)$$

$$= \sum_{D_{-i}} p(D_{-i}|D_i) \log \frac{p(D_{-i}|D_i)}{\widetilde{p}(D_{-i}|D_i)}$$

$$= D_{KL}(p\|\widetilde{p})$$

$$\geq \sum_{D_{-i}} \|p(D_{-i}|D_i) - \widetilde{p}(D_{-i}|D_i)\|^2.$$

We let the distribution difference vector be $\Delta_i$ (Note that here $\Delta_i$ is a $|\Theta|$-dimension vector), then we have

$$\Delta_{Rev_i'} \geq \sum_{D_{-i}} |p(D_{-i}|D_i) - \widetilde{p}(D_{-i}|D_i)|^2 \geq \sum_{D_{-i}} \left\| \sum_{\boldsymbol{\theta}} (p(\boldsymbol{\theta}|D_i) - \widetilde{p}(\boldsymbol{\theta}|D_i)) \cdot p(D_{-i}|\boldsymbol{\theta}) \right\|^2$$

$$= \|P_{-i}\Delta_i\|^2.$$

Since $e_i$ is the minimum singular value of $P_{-i}$ and thus $P_{-i}^T P_{-i} - e_i I$ is semi-positive, we have

$$\|P_{-i}\Delta_i\|^2 = \Delta_i^T P_{-i}^T P_{-i} \Delta_i$$

$$= \Delta_i^T (P_{-i}^T P_{-i} - e_i I)\Delta_i + \Delta_i^T e_i I \Delta_i$$

$$\geq \Delta_i^T e_i I \Delta_i$$

$$\geq e_i \Delta_i^T \Delta_i$$

$$= \|\Delta_i\| \cdot e_i.$$

Finally get the payment rule with revenue $Rev_i'$ is $e_i$-sensitive for agent $i$. If all $P_{-i}$ has rank $|\Theta|$, then all the singular values of the matrix $P_{-i}$ should have positive singular values and for all $i$, $e_i > 0$. By now we have proven that if all the $P_{-i}$ has rank $|\Theta|$, then the mechanism is sensitive. Since $p(\boldsymbol{\theta}|D_i) = p(D_i|\boldsymbol{\theta}) \cdot \frac{p(\boldsymbol{\theta})}{p(D_i)}$, we have the matrix equation:

$$Q_{-i} = \Lambda^{D_i^{-1}} \cdot P_{-i} \cdot \Lambda^{\boldsymbol{\theta}},$$

where $\Lambda^{D_i^{-1}} = \begin{bmatrix} \frac{1}{p(D_i^1)} & & & \\ & \frac{1}{p(D_i^2)} & & \\ & & \ddots & \\ & & & \frac{1}{p(D_i^{|\boldsymbol{\mathcal{D}}_i|})} \end{bmatrix}$ and $\Lambda^{\boldsymbol{\theta}} = \begin{bmatrix} p(\boldsymbol{\theta}_1) & & & \\ & p(\boldsymbol{\theta}_2) & & \\ & & \ddots & \\ & & & p(\boldsymbol{\theta}_{|\Theta|}) \end{bmatrix}.$

$p(D_i^j)$ is the probability that agent $i$ gets the dataset $D_i^j$. $p(\boldsymbol{\theta}_k)$ is the probability of the prior of the

parameter $\boldsymbol{\theta}$ with index $k$. Both are all diagnal matrices. Both of the diagnal matrices well-defined and full-rank. Thus the rank of $P_{-i}$ should be the same as $Q_{-i}$ and we have proved the first condition.

The proof for the second sufficient condition is directly derived from the paper [27] and the condition 1. We first define a matrix $G_i'$ with the same size as $G_i$ while its elements are $p(d_i|\boldsymbol{\theta})$ rather than $p(\boldsymbol{\theta}|d_i)$. Since for all $i' \in [n]$ the prediction matrix $P_{i'}$ is the columnwise Kronecker product (defined in Lemma 1 in [27] which is shown below) of $N_{i'}$ data generating matrices. By using the following Lemma in [27], if the k-rank of $G_{i'}'$ is $r$, then each time we multiply(columnwise Kronecker product) a matrix by $G_{i'}'$, the k-rank would increase by at least $rank_k(G_{i'}') - 1$, or reach the cap of $|\Theta|$.

**Lemma C.1.** *Consider two matrices* $\boldsymbol{A} = [\boldsymbol{a}_1, \boldsymbol{a}_2, \cdots, \boldsymbol{a}_F] \in \mathbb{R}^{I \times F}, \boldsymbol{B} = [\boldsymbol{b}_1, \boldsymbol{b}_2, \cdots, \boldsymbol{b}_F] \in \mathbb{R}^{J \times F}$ *and* $\boldsymbol{A} \odot_c \boldsymbol{B}$ *is the columnwise Krocnecker product of* $\boldsymbol{A}$ *and* $\boldsymbol{B}$ *defined as:*

$$\boldsymbol{A} \odot_c \boldsymbol{B} \triangleq [\boldsymbol{a}_1 \otimes \boldsymbol{b}_1, \boldsymbol{a}_2 \otimes \boldsymbol{b}_2, \cdots, \boldsymbol{a}_F \otimes \boldsymbol{b}_F],$$

*where* $\otimes$ *stands for the Kronecker product. It holds that*

$$rank_k(\boldsymbol{A} \odot_c \boldsymbol{B}) \geq \min\{rank_k(\boldsymbol{A}) + rank_k(\boldsymbol{B}) - 1, F\}.$$

Therefore the final k-rank of the $N_{i'}$ would be no less than $\min\{N_i \cdot (r - 1) + 1, |\Theta|\}$. We then need to calculate the k-rank of the out-prediction matrix of each agent $i$ and verify whether it is $|\Theta|$. Similarly, the out-prediction matrix of agent $i$ is the columnwise Kronecker product of all the other agent's prediction matrices. By the same lower bound tool in [27], the k-rank of $P_{-i}$ should be at least $\min\{\sum_{i' \neq i}(rank_k(G_{i'}') - 1) \cdot N_{i'} + 1, |\Theta|\}$ and by Theorem 5.2, if the k-rank of all prediction matrices are all $|\Theta|$, Mechanism 1 should be sensitive. $\square$

### C.5.2  Missing Proof for Theorem 5.3

When $\Theta \subseteq \mathbb{R}^m$ and a model in the exponential family is used, we prove that the mechanism will be sensitive if and only if for any $(\nu_i', \overline{\boldsymbol{\tau}}_i') \neq (\nu_i, \overline{\boldsymbol{\tau}}_i)$,

$$\Pr_{D_{-i}}\left[h_{D_{-i}}(\nu_i', \overline{\boldsymbol{\tau}}_i') \neq h_{D_{-i}}(\nu_i, \overline{\boldsymbol{\tau}}_i)\right] > 0. \tag{9}$$

We first show that the above condition is equivalent to that for any $(\nu_i', \overline{\boldsymbol{\tau}}_i') \neq (\nu_i, \overline{\boldsymbol{\tau}}_i)$,

$$\Pr_{D_{-i}|D_i}\left[h_{D_{-i}}(\nu_i', \overline{\boldsymbol{\tau}}_i') \neq h_{D_{-i}}(\nu_i, \overline{\boldsymbol{\tau}}_i)\right] > 0, \tag{10}$$

where $D_{-i}$ is drawn from $p(D_{-i}|D_i)$ but not $p(D_{-i})$. This is because, by conditional independence of the datasets, for any event $\mathcal{E}$, we have

$$\Pr_{D_{-i}|D_i}[\mathcal{E}] = \int_{\boldsymbol{\theta} \in \Theta} p(\boldsymbol{\theta}|D_i) \Pr_{D_{-i}|\boldsymbol{\theta}}[\mathcal{E}] d\boldsymbol{\theta}$$

and

$$\Pr_{D_{-i}}[\mathcal{E}] = \int_{\boldsymbol{\theta} \in \Theta} p(\boldsymbol{\theta}) \Pr_{D_{-i}|\boldsymbol{\theta}}[\mathcal{E}] d\boldsymbol{\theta}.$$

Since both $p(\boldsymbol{\theta})$ and $p(\boldsymbol{\theta}|D_i)$ are always positive because they are in exponential family, it should hold that

$$\Pr_{D_{-i}|D_i}[\mathcal{E}] > 0 \iff \Pr_{D_{-i}}[\mathcal{E}] > 0.$$

Therefore (9) is equivalent to (10), and we only need to show that the mechanism is sensitive if and only if (10) holds.

When we're using a (canonical) model in exponential family, the prior $p(\boldsymbol{\theta})$ and the posteriors $p(\boldsymbol{\theta}|D_i), p(\boldsymbol{\theta}|D_{-i})$ can be represented in the standard form (7),

$$p(\boldsymbol{\theta}) = \mathcal{P}(\boldsymbol{\theta}|\nu_0, \overline{\boldsymbol{\tau}}_0),$$
$$p(\boldsymbol{\theta}|D_i) = \mathcal{P}(\boldsymbol{\theta}|\nu_i, \overline{\boldsymbol{\tau}}_i),$$
$$p(\boldsymbol{\theta}|D_{-i}) = \mathcal{P}(\boldsymbol{\theta}|\nu_{-i}, \overline{\boldsymbol{\tau}}_{-i}),$$
$$p(\boldsymbol{\theta}|\widetilde{D}_i) = \mathcal{P}(\boldsymbol{\theta}|\nu_i', \overline{\boldsymbol{\tau}}_i'),$$

where $\nu_0, \overline{\boldsymbol{\tau}}_0$ are the parameters for the prior $p(\boldsymbol{\theta})$, $\nu_i, \overline{\boldsymbol{\tau}}_i$ are the parameters for the posterior $p(\boldsymbol{\theta}|D_i)$, $\nu_{-i}, \overline{\boldsymbol{\tau}}_{-i}$ are the parameters for the posterior $p(\boldsymbol{\theta}|D_{-i})$, and $\nu'_i, \overline{\boldsymbol{\tau}}'_i$ are the parameters for $p(\boldsymbol{\theta}|\widetilde{D}_i)$.

From the proof for Theorem 5.1, we know that the difference between the expected score of reporting $D_i$ and the expected score of reporting $\widetilde{D}_i \neq D_i$ is equal to

$$\Delta_{Rev} = D_{KL}(p(D_{-i}|D_i)\|p(D_{-i}|\widetilde{D}_i)).$$

Therefore if $p(D_{-i}|\widetilde{D}_i)$ differs from $p(D_{-i}|D_i)$ with non-zero probability, that is,

$$\Pr_{D_{-i}|D_i}[p(D_{-i}|D_i) \neq p(D_{-i}|\widetilde{D}_i)] > 0, \tag{11}$$

then $\Delta_{Rev} > 0$. By Lemma A.2 and Lemma A.3,

$$p(D_{-i}|D_i) = \int_{\boldsymbol{\theta} \in \Theta} \frac{p(\boldsymbol{\theta}|D_i)p(\boldsymbol{\theta}|D_{-i})}{p(\boldsymbol{\theta})} d\boldsymbol{\theta} = \frac{g(\nu_i, \overline{\boldsymbol{\tau}}_i)g(\nu_{-i}, \overline{\boldsymbol{\tau}}_{-i})}{g(\nu_0, \overline{\boldsymbol{\tau}}_0)g(\nu_i + \nu_{-i} - \nu_0, \frac{\nu_i \overline{\boldsymbol{\tau}}_i + \nu_{-i} \overline{\boldsymbol{\tau}}_{-i} - \nu_0 \overline{\boldsymbol{\tau}}_0}{\nu_i + \nu_{-i} - \nu_0})}.$$

$$p(D_{-i}|\widetilde{D}_i) = \int_{\boldsymbol{\theta} \in \Theta} \frac{p(\boldsymbol{\theta}|\widetilde{D}_i)p(\boldsymbol{\theta}|D_{-i})}{p(\boldsymbol{\theta})} d\boldsymbol{\theta} = \frac{g(\nu'_i, \overline{\boldsymbol{\tau}}'_i)g(\nu_{-i}, \overline{\boldsymbol{\tau}}_{-i})}{g(\nu_0, \overline{\boldsymbol{\tau}}_0)g(\nu'_i + \nu_{-i} - \nu_0, \frac{\nu'_i \overline{\boldsymbol{\tau}}'_i + \nu_{-i} \overline{\boldsymbol{\tau}}_{-i} - \nu_0 \overline{\boldsymbol{\tau}}_0}{\nu'_i + \nu_{-i} - \nu_0})}.$$

Therefore (11) is equivalent to

$$\Pr_{D_{-i}|D_i}[h_{D_{-i}}(\nu_i, \overline{\boldsymbol{\tau}}_i) \neq h_{D_{-i}}(\nu'_i, \overline{\boldsymbol{\tau}}'_i)] > 0.$$

Therefore if for all $(\nu'_i, \overline{\boldsymbol{\tau}}'_i) \neq (\nu_i, \overline{\boldsymbol{\tau}}_i)$, we have

$$\Pr_{D_{-i}|D_i}[h_{D_{-i}}(\nu_i, \overline{\boldsymbol{\tau}}_i) \neq h_{D_{-i}}(\nu'_i, \overline{\boldsymbol{\tau}}'_i)] > 0,$$

then reporting any $(\nu'_i, \overline{\boldsymbol{\tau}}'_i) \neq (\nu_i, \overline{\boldsymbol{\tau}}_i)$ will lead to a strictly lower expected score, which means the mechanism is sensitive. To prove the other direction, if the above condition does not hold, i.e., there exists $(\nu'_i, \overline{\boldsymbol{\tau}}'_i) \neq (\nu_i, \overline{\boldsymbol{\tau}}_i)$ with

$$\Pr_{D_{-i}|D_i}[h_{D_{-i}}(\nu'_i, \overline{\boldsymbol{\tau}}'_i) \neq h_{D_{-i}}(\nu_i, \overline{\boldsymbol{\tau}}_i)] = 0,$$

then reporting $(\nu'_i, \overline{\boldsymbol{\tau}}'_i) \neq (\nu_i, \overline{\boldsymbol{\tau}}_i)$ will give the same expected score as truthfully reporting $(\nu_i, \overline{\boldsymbol{\tau}}_i)$, which means that the mechanism is not sensitive.

## D  Multiple-time data acquisition

### D.1  Sensitivity analysis

We first give the sensitivity analysis for finite-size $|\Theta|$. The results are basically the same as the ones for the one-time data acquisition mechanism except that we do not give a lower bound for $\alpha$.

**Theorem D.1.** *When $|\Theta|$ is finite, if $f$ is strictly convex, then Mechanism 2 is sensitive in the first $T-1$ rounds if either of the following two conditions holds,*

*(1) $\forall i$, $Q_{-i}$ has rank $|\Theta|$.*

*(2) $\forall i$, $\sum_{i' \neq i}(rank_k(G_{i'}) - 1) \cdot N_{i'} + 1 \geq |\Theta|$.*

When $\Theta \subseteq \mathbb{R}^m$ is a continuous space, the results are entirely similar to the ones for Mechanism 1 but with slightly different proofs.

Suppose the data analyst uses a model from the exponential family so that the prior and all the posterior of $\boldsymbol{\theta}$ can be written in the form in Lemma 4.1. The sensitivity of the mechanism will depend on the normalization term $g(\nu, \overline{\boldsymbol{\tau}})$ (or equivalently, the partition function) of the pdf. Define

$$h_{D_{-i}}(\nu_i, \overline{\boldsymbol{\tau}}_i) = \frac{g(\nu_i, \overline{\boldsymbol{\tau}}_i)}{g(\nu_i + \nu_{-i} - \nu_0, \frac{\nu_i \overline{\boldsymbol{\tau}}_i + \nu_{-i} \overline{\boldsymbol{\tau}}_{-i} - \nu_0 \overline{\boldsymbol{\tau}}_0}{\nu_i + \nu_{-i} - \nu_0})}, \tag{12}$$

then we have the following sufficient and necessary conditions for the sensitivity of the mechanism.

**Theorem D.2.** *When $\Theta \subseteq \mathbb{R}^m$, if the data analyst uses a model in the exponential family and a strictly convex $f$, then Mechanism 2 is sensitive in the first $T-1$ rounds if and only if for any $(\nu'_i, \overline{\boldsymbol{\tau}}'_i) \neq (\nu_i, \overline{\boldsymbol{\tau}}_i)$, we have $\Pr_{D_{-i}}[h_{D_{-i}}(\nu'_i, \overline{\boldsymbol{\tau}}'_i) \neq h_{D_{-i}}(\nu_i, \overline{\boldsymbol{\tau}}_i)] > 0$.*

See Section 5 for interpretations of this theorem.

## D.2 Missing proofs

The following part are the proofs for our results.

**Proof of Theorem 6.1.** It is easy to verify that the mechanism is IR, budget feasible and symmetric. We prove the truthfulness as follows.

Let's look at the payment for day $t$. At day $t$, data provider $i$ reports a dataset $\widetilde{D}_i^{(t)}$. Assuming that all other data providers truthfully report $D_{-i}^{(t)}$, data provider $i$'s expected payment is decided by his expected score

$$\mathbb{E}_{(D_{-i}^{(t)}, D_{-i}^{(t+1)})|D_i^{(t)}}[s_i]$$

$$= \mathbb{E}_{D_{-i}^{(t+1)}} f'\left(\frac{1}{PMI(\widetilde{D}_i^{(t)}, D_{-i}^{(t+1)})}\right) - \mathbb{E}_{D_{-i}^{(t)}|D_i^{(t)}} f^*\left(f'\left(\frac{1}{PMI(\widetilde{D}_i^{(t)}, D_{-i}^{(t)})}\right)\right). \qquad (13)$$

The first expectation is taken over the marginal distribution $p(D_{-i}^{(t+1)})$ without conditioning on $D_i^{(t)}$ because $D^{(t+1)}$ is independent from $D^{(t)}$, so we have $p(D_{-i}^{(t+1)}|D_i^{(t)}) = p(D_{-i}^{(t+1)})$. Since the underlying distributions for different days are the same, we drop the superscripts for simplicity in the rest of the proof, so the expected score is written as

$$\mathbb{E}_{D_{-i}} f'\left(\frac{1}{PMI(\widetilde{D}_i, D_{-i})}\right) - \mathbb{E}_{D_{-i}|D_i} f^*\left(f'\left(\frac{1}{PMI(\widetilde{D}_i, D_{-i})}\right)\right). \qquad (14)$$

We then use Lemma 4.2 to get an upper bound of the expected score (14) and show that truthfully reporting $D_i$ achieves the upper bound. We apply Lemma 4.2 on two distributions of $D_{-i}$, the distribution of $D_{-i}$ conditioning on the observed $D_i$, $p(D_{-i}|D_i)$, and the marginal distribution $p(D_{-i})$. Then we get

$$D_f(p(D_{-i}|D_i), p(D_{-i})) \geq \sup_{g \in \mathcal{G}} \mathbb{E}_{D_{-i}}[g(D_{-i})] - \mathbb{E}_{D_{-i}|D_i}[f^*(g(D_{-i}))], \qquad (15)$$

where $f$ is the given convex function, $\mathcal{G}$ is the set of all real-valued functions of $D_{-i}$. The supremum is achieved and only achieved at function $g$ with

$$g(D_{-i}) = f'\left(\frac{p(D_{-i})}{p(D_{-i}|D_i)}\right) \text{ for all } D_{-i} \text{ with } p(D_{-i}|D_i) > 0. \qquad (16)$$

For a dataset $\widetilde{D}_i$, define function

$$g_{\widetilde{D}_i}(D_{-i}) = f'\left(\frac{1}{PMI(\widetilde{D}_i, D_{-i})}\right).$$

Then (15) gives an upper bound of the expected score (14) as

$$D_f(p(D_{-i}|D_i), p(D_{-i}))$$
$$\geq \mathbb{E}_{D_{-i}}\left[g_{\widetilde{D}_i}(D_{-i})\right] - \mathbb{E}_{D_{-i}|D_i}\left[f^*\left(g_{\widetilde{D}_i}(D_{-i})\right)\right]$$
$$= \mathbb{E}_{D_{-i}}\left[f'\left(\frac{1}{PMI(\widetilde{D}_i, D_{-i})}\right)\right] - \mathbb{E}_{D_{-i}|D_i}\left[f^*\left(f'\left(\frac{1}{PMI(\widetilde{D}_i, D_{-i})}\right)\right)\right]$$
$$= (14).$$

By (16), the upper bound is achieved only when

$$g_{\widetilde{D}_i}(D_{-i}) = f'\left(\frac{p(D_{-i})}{p(D_{-i}|D_i)}\right) \text{ for all } D_{-i} \text{ with } p(D_{-i}|D_i) > 0,$$

that is

$$f'\left(\frac{1}{PMI(\widetilde{D}_i, D_{-i})}\right) = f'\left(\frac{p(D_{-i})}{p(D_{-i}|D_i)}\right) \text{ for all } D_{-i} \text{ with } p(D_{-i}|D_i) > 0. \qquad (17)$$

Then it is easy to prove the truthfulness. Truthfully reporting $D_i$ achieves (17) because by Lemma A.2, for all $D_i$ and $D_{-i}$,

$$PMI(D_i, D_{-i}) = \frac{p(D_i, D_{-i})}{p(D_i)p(D_{-i})} = \frac{p(D_{-i}|D_i)}{p(D_{-i})}.$$

Again, let $\boldsymbol{Q_{-i}}$ be a $(\Pi_{j \in [n], j \neq i} |\mathcal{D}_j|^{N_j}) \times |\Theta|$ matrix with elements equal to $p(\boldsymbol{\theta}|D_{-i})$ and let $G_i$ be the $|\mathcal{D}_i| \times |\Theta|$ data generating matrix with elements equal to $p(\boldsymbol{\theta}|d_i)$. Then we have the following sufficient conditions for the mechanism's sensitivity.

**Proof of Theorem D.1.** We then prove the sensitivity. For discrete and finite-size $\Theta$, we prove that when $f$ is strictly convex and $\boldsymbol{Q_{-i}}$ has rank $|\Theta|$, the mechanism is sensitive. When $f$ is strictly convex, $f'$ is a strictly increasing function. Let $\widetilde{\mathbf{q}}_i = p(\boldsymbol{\theta}|\widetilde{D}_i)$. Then accordint to the definition of $PMI(\cdot)$, condition (17) is equivalent to

$$PMI(\widetilde{D}_i, D_{-i}) = \sum_{\boldsymbol{\theta} \in \Theta} \frac{\widetilde{\mathbf{q}}_i \cdot p(\boldsymbol{\theta}|D_{-i})}{p(\boldsymbol{\theta})} = \frac{p(D_{-i}|D_i)}{p(D_{-i})} \quad \text{for all } D_{-i} \text{ with } p(D_{-i}|D_i) > 0. \quad (18)$$

We show that when matrix $\boldsymbol{Q_{-i}}$ has rank $|\Theta|$, $\widetilde{\mathbf{q}}_i = p(\boldsymbol{\theta}|D_i)$ is the only solution of (18), which means that the payment rule is sensitive. Then suppose $\widetilde{\mathbf{q}}_i = p(\boldsymbol{\theta}|D_i)$ and $\widetilde{\mathbf{q}}_i = p(\boldsymbol{\theta}|\widetilde{D}_i)$ are both solutions of (18), then we should have

$$p(D_{-i}|\widetilde{D}_i) = p(D_{-i}|D_i) \text{ for all } D_{-i} \text{ with } p(D_{-i}|D_i) > 0.$$

In addition, because

$$\sum_{D_{-i}} p(D_{-i}|\widetilde{D}_i) = 1 = \sum_{D_{-i}} p(D_{-i}|D_i)$$

and $p(D_{-i}|\widetilde{D}_i) \geq 0$, we must also have $p(D_{-i}|\widetilde{D}_i) = 0$ for all $D_{-i}$ with $p(D_{-i}|D_i) = 0$. Therefore we have

$$PMI(\widetilde{D}_i, D_{-i}) = PMI(D_i, D_{-i}) \text{ for all } D_{-i}.$$

Since $PMI(\cdot)$ can be written as,

$$PMI(\widetilde{D}_i, D_{-i}) = \sum_{\boldsymbol{\theta} \in \Theta} \frac{p(\boldsymbol{\theta}|\widetilde{D}_i)p(\boldsymbol{\theta}|D_{-i})}{p(\boldsymbol{\theta})} = (\boldsymbol{Q_{-i}}\boldsymbol{\Lambda}\widetilde{\mathbf{q}}_i)_{D_{-i}}$$

where $\boldsymbol{\Lambda}$ is the $|\Theta| \times |\Theta|$ diagonal matrix with $1/p(\boldsymbol{\theta})$ on the diagonal. So we have

$$\boldsymbol{Q_{-i}}\boldsymbol{\Lambda}p(\boldsymbol{\theta}|D_i) = \boldsymbol{Q_{-i}}\boldsymbol{\Lambda}\boldsymbol{q} \quad \Longrightarrow \quad \boldsymbol{Q_{-i}}\boldsymbol{\Lambda}(p(\boldsymbol{\theta}|D_i) - \boldsymbol{q}) = 0.$$

Since $\boldsymbol{Q_{-i}}\boldsymbol{\Lambda}$ must have rank $|\Theta|$, which means that the columns of $\boldsymbol{Q_{-i}}\boldsymbol{\Lambda}$ are linearly independent, we must have

$$p(\boldsymbol{\theta}|D_i) - \boldsymbol{q} = 0,$$

which completes our proof of sensitivity for finite-size $\Theta$. The proof of condition (2) is the same as the proof of Theorem C.3 condition (2).

**Proof of Theorem D.2.** When $\Theta \subseteq \mathbb{R}^m$ and a model in the exponential family is used, we prove that when $f$ is strictly convex, the mechanism will be sensitive if and only if for any $(\nu_i', \overline{\boldsymbol{\tau}}_i') \neq (\nu_i, \overline{\boldsymbol{\tau}}_i)$,

$$\Pr_{D_{-i}}[h_{D_{-i}}(\nu_i', \overline{\boldsymbol{\tau}}_i') \neq h_{D_{-i}}(\nu_i, \overline{\boldsymbol{\tau}}_i)] > 0. \quad (19)$$

We first show that the above condition is equivalent to that for any $(\nu_i', \overline{\boldsymbol{\tau}}_i') \neq (\nu_i, \overline{\boldsymbol{\tau}}_i)$,

$$\Pr_{D_{-i}|D_i}[h_{D_{-i}}(\nu_i', \overline{\boldsymbol{\tau}}_i') \neq h_{D_{-i}}(\nu_i, \overline{\boldsymbol{\tau}}_i)] > 0, \quad (20)$$

where $D_{-i}$ is drawn from $p(D_{-i}|D_i)$ but not $p(D_{-i})$. This is because, by conditional independence of the datasets, for any event $\mathcal{E}$, we have

$$\Pr_{D_{-i}|D_i}[\mathcal{E}] = \int_{\boldsymbol{\theta} \in \Theta} p(\boldsymbol{\theta}|D_i) \Pr_{D_{-i}|\boldsymbol{\theta}}[\mathcal{E}] d\boldsymbol{\theta}$$

and

$$\Pr_{D_{-i}}[\mathcal{E}] = \int_{\boldsymbol{\theta} \in \Theta} p(\boldsymbol{\theta}) \Pr_{D_{-i}|\boldsymbol{\theta}}[\mathcal{E}] \, d\boldsymbol{\theta}.$$

Since both $p(\boldsymbol{\theta})$ and $p(\boldsymbol{\theta}|D_i)$ are always positive because they are in exponential family, it should hold that

$$\Pr_{D_{-i}|D_i}[\mathcal{E}] > 0 \iff \Pr_{D_{-i}}[\mathcal{E}] > 0.$$

Therefore (19) is equivalent to (20), and we only need to show that the mechanism is sensitive if and only if (20) holds.

Let $\widetilde{\mathbf{q}}_i = p(\boldsymbol{\theta}|\widetilde{D}_i)$. We then again apply Lemma 4.2. By Lemma 4.2 and the strict convexity of $f$, $\widetilde{\mathbf{q}}_i$ achieves the supremum if and only if

$$PMI(\widetilde{D}_i, D_{-i}) = \frac{p(D_{-i}|D_i)}{p(D_{-i})} \quad \text{for all } D_{-i} \text{ with } p(D_{-i}|D_i) > 0.$$

By the definition of $PMI$ and Lemma A.2, the above condition is equivalent to

$$\int_{\boldsymbol{\theta} \in \Theta} \frac{\widetilde{\mathbf{q}}_i(\boldsymbol{\theta}) p(\boldsymbol{\theta}|D_{-i})}{p(\boldsymbol{\theta})} \, d\boldsymbol{\theta} = \int_{\boldsymbol{\theta} \in \Theta} \frac{p(\boldsymbol{\theta}|D_i) p(\boldsymbol{\theta}|D_{-i})}{p(\boldsymbol{\theta})} \, d\boldsymbol{\theta} \quad \text{for all } D_{-i} \text{ with } p(D_{-i}|D_i) > 0. \quad (21)$$

When we're using a (canonical) model in exponential family, the prior $p(\boldsymbol{\theta})$ and the posteriors $p(\boldsymbol{\theta}|D_i), p(\boldsymbol{\theta}|D_{-i})$ can be represented in the standard form (7),

$$p(\boldsymbol{\theta}) = \mathcal{P}(\boldsymbol{\theta}|\nu_0, \overline{\boldsymbol{\tau}}_0),$$
$$p(\boldsymbol{\theta}|D_i) = \mathcal{P}(\boldsymbol{\theta}|\nu_i, \overline{\boldsymbol{\tau}}_i),$$
$$p(\boldsymbol{\theta}|D_{-i}) = \mathcal{P}(\boldsymbol{\theta}|\nu_{-i}, \overline{\boldsymbol{\tau}}_{-i}),$$
$$\widetilde{\mathbf{q}}_i = \mathcal{P}(\boldsymbol{\theta}|\nu_i', \overline{\boldsymbol{\tau}}_i'),$$

where $\nu_0, \overline{\boldsymbol{\tau}}_0$ are the parameters for the prior $p(\boldsymbol{\theta})$, $\nu_i, \overline{\boldsymbol{\tau}}_i$ are the parameters for the posterior $p(\boldsymbol{\theta}|D_i)$, $\nu_{-i}, \overline{\boldsymbol{\tau}}_{-i}$ are the parameters for the posterior $p(\boldsymbol{\theta}|D_{-i})$, and $\nu_i', \overline{\boldsymbol{\tau}}_i'$ are the parameters for $\widetilde{\mathbf{q}}_i$. Then by Lemma A.3, the condition that $\widetilde{\mathbf{q}}_i$ achieves the supremum (21) is equivalent to

$$\frac{g(\nu_i', \overline{\boldsymbol{\tau}}_i')}{g(\nu_i' + \nu_{-i} - \nu_0, \frac{\nu_i' \overline{\boldsymbol{\tau}}_i' + \nu_{-i} \overline{\boldsymbol{\tau}}_{-i} - \nu_0 \overline{\boldsymbol{\tau}}_0}{\nu_i' + \nu_{-i} - \nu_0})} = \frac{g(\nu_i, \overline{\boldsymbol{\tau}}_i)}{g(\nu_i + \nu_{-i} - \nu_0, \frac{\nu_i \overline{\boldsymbol{\tau}}_i + \nu_{-i} \overline{\boldsymbol{\tau}}_{-i} - \nu_0 \overline{\boldsymbol{\tau}}_0}{\nu_i + \nu_{-i} - \nu_0})}. \quad (22)$$

which, by our definition of $h(\cdot)$, is just

$$h_{D_{-i}}(\nu_i', \overline{\boldsymbol{\tau}}_i') = h_{D_{-i}}(\nu_i, \overline{\boldsymbol{\tau}}_i), \quad \text{for all } D_{-i} \text{ with } p(D_{-i}|D_i) > 0.$$

Now we are ready to prove Theorem D.2. Since (19) is equivalent to (20), we only need to show that the mechanism is sensitive if and only if for all $(\nu_i', \overline{\boldsymbol{\tau}}_i') \neq (\nu_i, \overline{\boldsymbol{\tau}}_i)$,

$$\Pr_{D_{-i}|D_i}[h_{D_{-i}}(\nu_i', \overline{\boldsymbol{\tau}}_i') \neq h_{D_{-i}}(\nu_i, \overline{\boldsymbol{\tau}}_i)] > 0.$$

If the above condition holds, then $\widetilde{\mathbf{q}}_i$ with parameters $(\nu_i', \overline{\boldsymbol{\tau}}_i') \neq (\nu_i, \overline{\boldsymbol{\tau}}_i)$ should have a non-zero loss in the expected score (14) compared to the optimal solution $p(\boldsymbol{\theta}|D_i)$ with parameters $(\nu_i, \overline{\boldsymbol{\tau}}_i)$, which means that the mechanism is sensitive. For the other direction, if the condition does not hold, i.e., there exists $(\nu_i', \overline{\boldsymbol{\tau}}_i') \neq (\nu_i, \overline{\boldsymbol{\tau}}_i)$ with

$$\Pr_{D_{-i}|D_i}[h_{D_{-i}}(\nu_i', \overline{\boldsymbol{\tau}}_i') \neq h_{D_{-i}}(\nu_i, \overline{\boldsymbol{\tau}}_i)] = 0,$$

then reporting $(\nu_i', \overline{\boldsymbol{\tau}}_i') \neq (\nu_i, \overline{\boldsymbol{\tau}}_i)$ will give the same expected score as truthfully reporting $(\nu_i, \overline{\boldsymbol{\tau}}_i)$, which means that the mechanism is not sensitive.