[Reviews · NeurIPS 2020]

Review 1

Summary and Contributions: Please see the comments to the authors section

Strengths: Please see the comments to the authors section

Weaknesses: Please see the comments to the authors section

Correctness: Please see the comments to the authors section

Clarity: Please see the comments to the authors section

Relation to Prior Work: Please see the comments to the authors section

Reproducibility: Yes

Additional Feedback: The paper studies the problem of designing payment rules for acquiring data. The problem here is that the procurer has no way of directly verifying the accuracy of the data procured. The goal is to design a payment scheme for the procurer so that when he announces that scheme, the data providers don't feel incentivized to misreport their data. The problem as such is very close the peer prediction literature which suffers from the same problem of being unable to directly verify accuracy. The peer prediction literature has seen a lot of activity recently, and one of the main ideas from there is to pay based on the pointwise-mutual-information (PMI) of two datasets. However directly applying the log-PMI rule developed previously runs into problems of budget infeasibility, non-individual-rationality. The paper adds a normalization to the log-PMI score suggested by prior work --- the normalization is not trivial, but not too involved either. The paper argues that the mechanism is not just truthful, but also that truthful is essentially the only utility maximizing strategy, something they formalize as sensitivity. The paper also studies the repeated setting case where the data is repeatedly generated and applies the same normalization idea. The paper is very well written, both the introduction and technical section. Comparison to prior work is quite clear. The degree of technical advance compared to prior work may not be sufficient for NeurIPS --- this is a paper that I would like to see published, but maybe doesn't yet clear the NeurIPS bar. [After author rebuttal]: Thanks for the response. I read it and updated my score.


Review 2

Summary and Contributions: This paper examines the problem of incentivizing the truthful revelation of data for machine learning. The work builds heavily on prior work [14,15] which establishes a way to do so via peer prediction. In doing so it makes three contributions. First, a way to transform the payments is given so that they are not just truthful but also IR, budget-fixed, and symmetric if the distribution is such that the range of possible payments can be bounded. Second, the transformed payments are shown to satisfy a notion of sensitivity, which means that truthful reporting is strictly better than any misreport that damages the learning from the combined data sets. Third, a trick is introduced to use repeated elicitation to eliminate the requirement that bounds on the range of payments can be derived from the distribution.

Strengths: While the first contribution seems to rely on standard techniques, the second and third contributions seem to be quite novel and interesting. The notion of sensitivity seems a natural desideratum analogous to the concept of strict properness in the scoring rules literature. The third seems a useful relaxation of the technical requirement as well as possibly of interest in other settings where mechanisms like PMI can be applied.

Weaknesses: The core technical content in Sections 5 and 6 is quite compressed, resulting in an exposition that is hard to follow, as discussed under clarity below. I think this rises to the level that it significantly weakens the paper, because while there is enough to convince me that the contributions seem interesting, the presentation is so compressed that I have a hard time digesting what they actually say.

Correctness: Apart from minor presentation issues mentioned under additional feedback the results appear correct.

Clarity: The material from line 212 to the end of Section 5 gives a precise explanation of what the technical conditions are to guarantee sensitivity, but provides very little in the way of intuition for either why those conditions suffice or what those conditions mean. As a result, I don’t even have a clear picture of how common it is for a model to satisfy these conditions. For the material in Section 6, I think very little of what is going on has anything to do with the data-sets or the mechanism used and instead relies entirely on what ought to be a technical lemma about equation (5) that is instead buried in the proof of Theorem 6.1. I’d like to see the relevant lemma factored out and stated on its own since it may be useful in other settings and doing so will provide more intuition about why the Theorem is actually true than the brief discussion on ines 267-269 that I can’t really follow. Also, because of the way it is written I can’t actually tell whether this lemma is a novel contribution or not. If it is already found in previous work I would significantly reduce my enthusiasm for the third contribution.

Relation to Prior Work: Generally the coverage of prior work is good. One omission is that there is a small literature on aggregating datasets using scoring rules (rather than peer prediction) that uses the same sorts of modeling in terms of exponential families. See the below and subsequent work. @article{fang2007putting, title={" Putting Your Money Where Your Mouth Is"-A Betting Platform for Better Prediction}, author={Fang, Fang and Stinchcombe, Maxwell and Whinston, Andrew}, journal={Review of Network Economics}, volume={6}, number={2}, year={2007}, publisher={De Gruyter} }

Reproducibility: Yes

Additional Feedback: 84 – Here since you are talking about a particular way of using \theta as a parameterization I think it should be “an exponential family” rather than “the exponential family” 125 – The definition of sensitivity is quite hard to parse. I think it could be simplified using the idea that, since the mechanism only actually computes scores based on p(\theta | D), the mechanism could just as easily be defined as taking that as the report directly rather than D, at which point this reduces to more standard notions of strict propriety. The language of property elicitation, which has recently found other use in peer prediction, may be helpful for reformulating in this fashion. @article{frongillo2017geometric, title={A geometric perspective on minimal peer prediction}, author={Frongillo, Rafael and Witkowski, Jens}, journal={ACM Transactions on Economics and Computation (TEAC)}, volume={5}, number={3}, pages={1--27}, year={2017}, publisher={ACM New York, NY, USA} } 159 – In the definitions here, there is a notation conflict, as x is used as a generic stand-in while previously it is more specifically used for the features. 211 – The proof of Theorem 5.1 in the supplemental material only seems to state and show the weaker condition that the mechanism is budget bounded, although the argument for budget-fixed seems straightforward. Post response: Thank you for the response.


Review 3

Summary and Contributions: This paper considers the problem of paying users for data when the mechanism has no ground truth to test against. It designs two mechanisms for this. One that works with just one batch of data, and another that works if new data arrives over time. The technical heart is borrowed from the peer prediction literature. However, they also make it work for the exponential family of mechanisms and ensure the mechanisms are budget balanced and IR.

Strengths: + Innovative contribution for an very interesting problem. + Extends previous peer prediction results in interesting directions.

Weaknesses: - While the results are nice, there is not too much technical novelty here. -Some more proof-reading/putting through Grammery required.

Correctness: I did not check everything carefully, but nothing seems incorrect.

Clarity: a few typoes; it is pretty well written overall.

Relation to Prior Work: yes

Reproducibility: Yes

Additional Feedback: Summary: Overall, I think this is a nice contribution for this conference. It has an interesting take on an important problem. While the technical novelty is limited; there is still work and value in generalizing the previous techniques to this new context. Other comments: In example 3.2 it seems like you need p(theta) as well to compute p(\theta|x_i, y_i) not just p(\x_i, y_i|theta). In fact, I don’t really understand this claim as stated. It seems more a claim about computation rather than knowledge. Once you can compute p(\x_i, y_i|theta) and know p(theta), then you know the entire prior. But this does not mean that it is easy to compute properties you desire about it. The function g should be defined (better) in definition 4.2. Typoes: Line 54 (remove “to”); Line 78; Line 96; Line 155 -> “Preliminaries”; ******AFTER REBUTTAL************ The rebuttal was not addressed to me, and after reading it my review remains unchanged.


Review 4

Summary and Contributions: This paper studies the following problem. There are n data providers; provider i holds a dataset D_i. There is some joint distribution p( \theta, D_1, \dots, D_n ). There is a known prior p(\theta), and the distributions are independent conditioned on \theta. The goal is to incentivize the providers to report their true datasets (using a budget of at most B). The proposed mechanism is based on the pointwise information. Specifically, each provider is assigned a score based on the pointwise information of the reported dataset and sets a payment proportional to this score. This mechanism is shown to be truthful and, under mild assumptions, to be “sensitive”: it is strictly better to report a dataset that induces a correct posterior p( \theta | D_i ). Similar ideas work when the principal needs to acquire data repeatedly.

Strengths: The mechanisms provided are fairly clean and have strong theoretical guarantees. Specifically, I was initially surprised that getting a sensitive mechanism is even possible. In terms of fit, this paper is relevant to the NeurIPS community.

Weaknesses: There seems to be a lot of overlap with the related work in terms of techniques. So, if this paper is seen as an application of known techniques to a new problem then the contribution is marginal.

Correctness: Claims seem to be correct.

Clarity: Overall writing is good.

Relation to Prior Work: The authors are honest about the overlap in techniques with the related work, and the related papers are sufficiently discussed.

Reproducibility: Yes

Additional Feedback: By truthfulness here it is meant bayesian incentive compatibility. The term "truthful" typically means "DSIC", so it would be useful to stress that this is not the case here. Update: After reading the author rebuttal and the discussion I am more positive towards this paper. I updated my score to reflect this.

[Author Response · NeurIPS 2020]

We thank all the reviewers for their feedback!

Our paper formalizes a data acquisition problem when one cannot verify the true labels of the collected data. Under-
standing this problem is important as high-quality data are crucial for high-quality output of machine learning systems.
The writing of our submission focused more on the basics to ensure clarity for the general, diverse Neurips readers.
Most of the technical results were either deferred to the appendix or compressed to fit in the page limit. We thus want to
elaborate a bit more on some of our technical contributions.

One of our major technical contributions is the explicit sensitivity guarantee of the peer-prediction style mechanisms.
Sensitivity is the key property we want for a data acquisition mechanism, which prevents data providers from harmful
misreporting. In this work, we provide explicit and verifiable conditions for sensitivity, which is absent in the previous
work. In [1], checking whether the mechanism will be sensitive requires knowing whether a system of linear equations
(which has exponential size in our problem) has a unique solution. So it is not clear how likely the mechanisms will be
sensitive. In our data acquisition setting, we are able to give much stronger guarantees:

1. When $\theta$ has a finite support, we give a sufficient condition for sensitivity. The condition only uses the
distribution of $\theta$ and a single data point (Corollary 5.1). The basic message is that when there is enough
"correlation" between other people's data and $\theta$, the mechanism will be sensitive. Corollary 5.1 quantifies the
"correlation" using the k-rank of the distribution matrix. It is arguably not difficult to have enough "correlation":
a naive relaxation of Corollary 5.1 says that assuming different $\theta$ lead to different data distributions, the
mechanism will be sensitive if the total number of other people's data points $\geq |\Theta| - 1$.[1]

2. For an exponential family distribution, we give the explicit necessary and sufficient condition for the mechanism
to be sensitive (Theorem 5.3), which is based on a function of the normalization term of the exponential family
probability.

This kind of sensitivity guarantee is possible because of the special structure of the reports (or the signals): each dataset
consists of i.i.d. samples (conditioning on theta), despite the fact that the signal space for our problem is much larger
(the number of possible realizations of a dataset is exponentially large).

Besides sensitivity, another important property of our mechanisms is budget feasibility.

1. For the one-time acquisition, the log PMI payment rule requires PMI to be bounded. We give a polynomial-time
method (a trivial method would take exponential time) to find the bounds for finite-size $\Theta$ (Appendix C.2).

2. For the multiple-time mechanism, budget feasibility can be guaranteed for any underlying distribution. This is
achieved by carefully choosing the convex function $f$ in the $f$-mutual information gain to have a bounded
derivative.

If the paper is accepted, we will add more explanation about our technical results and factor out some omitted key steps
in the proofs. We thank all the comments on the writing/related work/typos etc. We will revise the paper accordingly.

**Other minor comments:**

**About the proof of Theorem 5.1.**  Yes, showing budget-fixed from budget bounded is straightforward based on our
normalization. We will make it clear in the final version. Thanks for pointing this out.

**About Example 3.2.**  It is a claim about knowledge: we may not need to know the entire prior to compute $p(\theta|\mathbf{x}_i, y_i)$.
In this example, it is right we need to know $p(\theta)$, but we do not necessarily need $p(\mathbf{x}_i, y_i|\theta)$ because $p(y_i|\mathbf{x}_i, \theta)$
will be sufficient.

## Footnotes

[1]Here we have found a typo in our statement of Corollary 5.1, the ineuqality should be $\sum_{j \neq i}(rank_k(G_j) - 1) N_j + 1 \geq |\Theta|$

# References

[1] Yuqing Kong and Grant Schoenebeck. Water from two rocks: Maximizing the mutual information. In *Proceedings*
*of the 2018 ACM Conference on Economics and Computation*, pages 177–194, 2018.

rather than $\sum_{j \neq i} rank_k(G_j) (N_j - 1) + 1 \geq |\Theta|$. We will correct this in our final version.

[Meta-Review · NeurIPS 2020]

This paper is quite strong and has nice ideas; it is also well-written. The reviewers agreed that this would make a positive contribution to NeurIPS this year, and we would like to see it accepted. But please take care to address the more detailed comments from the reviewers when preparing your final camera-ready.